# Alternative splicing at neuroligin site A regulates glycan interaction and synaptogenic activity

**Shinichiro Oku[1], Huijuan Feng[2], Steven Connor[1,3], Andrea Toledo[4], Peng Zhang[1†], Yue Zhang[1], Olivier Thoumine[4], Chaolin Zhang[2], Ann Marie Craig[1]\***

[1]Djavad Mowafaghian Centre for Brain Health and Department of Psychiatry, University of British Columbia, Vancouver, Canada; [2]Departments of Systems Biology and Biochemistry and Molecular Biophysics, Center for Motor Neuron Biology and Disease, Columbia University, New York, United States; [3]Department of Biology, York University, Toronto, Canada; [4]Interdisciplinary Institute for Neuroscience UMR 5297, CNRS and University of Bordeaux, Bordeaux, France

**Abstract** Post-transcriptional mechanisms regulating cell surface synaptic organizing complexes that control the properties of connections in brain circuits are poorly understood. Alternative splicing regulates the prototypical synaptic organizing complex, neuroligin-neurexin. In contrast to the well-studied neuroligin splice site B, little is known about splice site A. We discovered that inclusion of the positively charged A1 insert in mouse neuroligin-1 increases its binding to heparan sulphate, a modification on neurexin. The A1 insert increases neurexin recruitment, presynaptic differentiation, and synaptic transmission mediated by neuroligin-1. We propose that the A1 insert could be a target for alleviating the consequences of deleterious *NLGN1/3* mutations, supported by assays with the autism-linked neuroligin-1-P89L mutant. An enrichment of neuroligin-1 A1 in GABAergic neuron types suggests a role in synchrony of cortical circuits. Altogether, these data reveal an unusual mode by which neuroligin splicing controls synapse development through protein-glycan interaction and identify it as a potential therapeutic target.

**\*For correspondence:** acraig@mail.ubc.ca

**Present address:** †Department of Neurosciences, Case Western Reserve University, Cleveland, United States

**Competing interests:** The authors declare that no competing interests exist.

## Introduction

Synapses are specialized sites of intercellular communication where chemical transmission occurs between neurons in brain circuits. Cell surface trans-synaptic organizing complexes recruit presynaptic vesicles and the neurotransmitter release machinery and apposing postsynaptic receptors and scaffold proteins. Dendritic neuroligins (NLGNs) and their axonal neurexin (NRXN) binding partners form a prototypical synaptic organizing complex (*Südhof, 2017*). NLGNs bind to two interfaces on NRXN, to the laminin-neurexin-sex hormone (LNS) protein domain and to the heparan sulphate (HS) glycan chain (*Zhang et al., 2018*). Mutations in all *NLGN* and *NRXN* genes are associated with neuropsychiatric disorders. NLGNs and NRXNs are evolutionarily conserved and essential for mouse survival (*Missler et al., 2003*; *Varoqueaux et al., 2006*). NLGNs have additional extracellular domain binding partners, cell surface MDGAs, also linked to neuropsychiatric disorders (*Connor et al., 2019*). MDGA binding to NLGNs occludes the NRXN binding site, thus preventing NLGN-NRXN interaction and suppressing synapse development. Intracellularly, NLGNs can bind to the excitatory postsynaptic scaffold PSD-95 and the inhibitory postsynaptic scaffold gephyrin, interactions that are regulated by phosphorylation (*Jeong et al., 2019*; *Jeong et al., 2017*; *Letellier et al., 2020*; *Letellier et al., 2018*). NLGN1 functions at excitatory glutamatergic synapses, NLGN2 at inhibitory GABAergic synapses, and NLGN3 at both.

NLGN1 is essential for normal NMDA receptor recruitment, long-term potentiation, and spatial learning and memory (*Blundell et al., 2010*; *Budreck et al., 2013*; *Jiang et al., 2017*; *Shipman and Nicoll, 2012*). Additional competitive effects of NLGN1 on synapse number were revealed with sparse knockdown (*Kwon et al., 2012*) and roles in long-term depression revealed in *Nlgn1* heterozygous mice (*Dang et al., 2018*). Copy number, truncating, and missense variants in *NLGN1* are associated with autism spectrum disorders (ASD) (*Glessner et al., 2009*; *Nakanishi et al., 2017*; *O'Roak et al., 2012*; *Tejada et al., 2019*) and obsessive-compulsive disorder (*Gazzellone et al., 2016*) and *NLGN1* polymorphisms are linked to schizophrenia (*Chen et al., 2018*; *Zhang et al., 2015*).

NLGNs are regulated by alternative splicing (see *Figure 1—figure supplement 1* for gene structures of NLGN1-3). The function of splice site B present only in NLGN1 is well understood. The B insert blocks the interaction of NLGN1 with α-NRXNs, reduces binding to β-NRXNs, and reduces binding to MDGAs (*Boucard et al., 2005*; *Elegheert et al., 2017*; *Koehnke et al., 2010*). The major form of NLGN1 contains this B insert but lacks an insert at splice site A (*Chih et al., 2006*). Additional forms of NLGN1 and NLGN3 can have conserved A1, A2 or A1A2 splice inserts whereas NLGN2, NLGN4, and NLGN5 can have a single A insert similar to A2 (*Bolliger et al., 2008*). However, the NLGN A inserts have little effect on interaction with NRXN or MDGA protein domains (*Comoletti et al., 2006*; *Elegheert et al., 2017*; *Koehnke et al., 2010*). At the cellular level, the A2 insert promotes association with GABAergic synapses only in the absence of the B insert (*Chih et al., 2006*). To the best of our knowledge, a function for the NLGN1 A1 insert has not been reported. Here, we show that the A1 insert increases the binding of NLGN1 to HS, a modification on NRXN, increasing NLGN1 synaptogenic activity. We further report cell type regulation of A1 insertion, identify Rbfox1-3 as potential regulators, and show that A1 insertion can partially restore function of an ASD-linked missense variant of NLGN1.

## Results

### The NLGN1 A1 insert increases heparin binding

The A1 insert of NLGN1 and NLGN3 forms a positively charged surface, with 9 of the 20 amino acids being arginine or lysine (*Figure 1A–C*). The two internal cysteine residues form a disulfide bond (*Hoffman et al., 2004*), contributing to the extended surface. Another constitutive positively charged surface on NLGNs near the dimer interface binds the HS chain of NRXN, an interaction that is necessary along with protein-protein interactions for full NLGN-NRXN complex formation and function (*Zhang et al., 2018*). We wondered whether the A1 insert with its high density of positively charged surface residues might participate in binding of NLGN to the HS chain of NRXN. To test this idea, we compared relative binding affinity for different splice variants of recombinant purified NLGN1 ectodomain to a heparin column. Indeed, NLGN1 variants containing the A1 insert required higher concentrations of salt than variants lacking A1 for elution from heparin (*Figure 1D*; all NLGN1 variants expressed in this study contain the B insert). This indicates that A1 insertion increases the affinity of NLGN1 for HS. We further wondered whether the A2 insert with its high acidic composition might counteract the effect of A1 but we observed only a small shift in the peak of elution between NLGN1 +A1 and NLGN1 +A1A2.

### The NLGN1 A1 insert enhances neurexin recruitment in coculture

To assess the relative interaction of NLGN1 A splice variants with NRXN, we used a cell-based recruitment assay. NLGN1-expressing non-neuronal cells cocultured with neurons recruit axonal NRXN to contact sites (*Zhang et al., 2018*). If the observed differences in heparin binding among NLGN splice variants reflect differences in binding to native HS modified NRXN, we expect NLGN1 A1-containing splice variants to recruit more NRXN than NLGN1 variants lacking A1. Indeed, we found that Myc-tagged NLGN1 +A1 and +A1A2 splice forms expressed on COS7 cells cocultured with rat hippocampal neurons recruited more native neuronal NRXN to contact sites than NLGN1 -A or +A2 (*Figure 2A–C*). We used this cell-based assay to assess NRXN-NLGN1 interaction because recombinant NRXN produced in HEK293 cells is poorly HS modified (*Zhang et al., 2018*), making it difficult to do direct binding assays with appropriately glycosylated NRXN. Three studies used surface plasmon resonance to assess the effect of NLGN splicing on binding affinity to a truncated

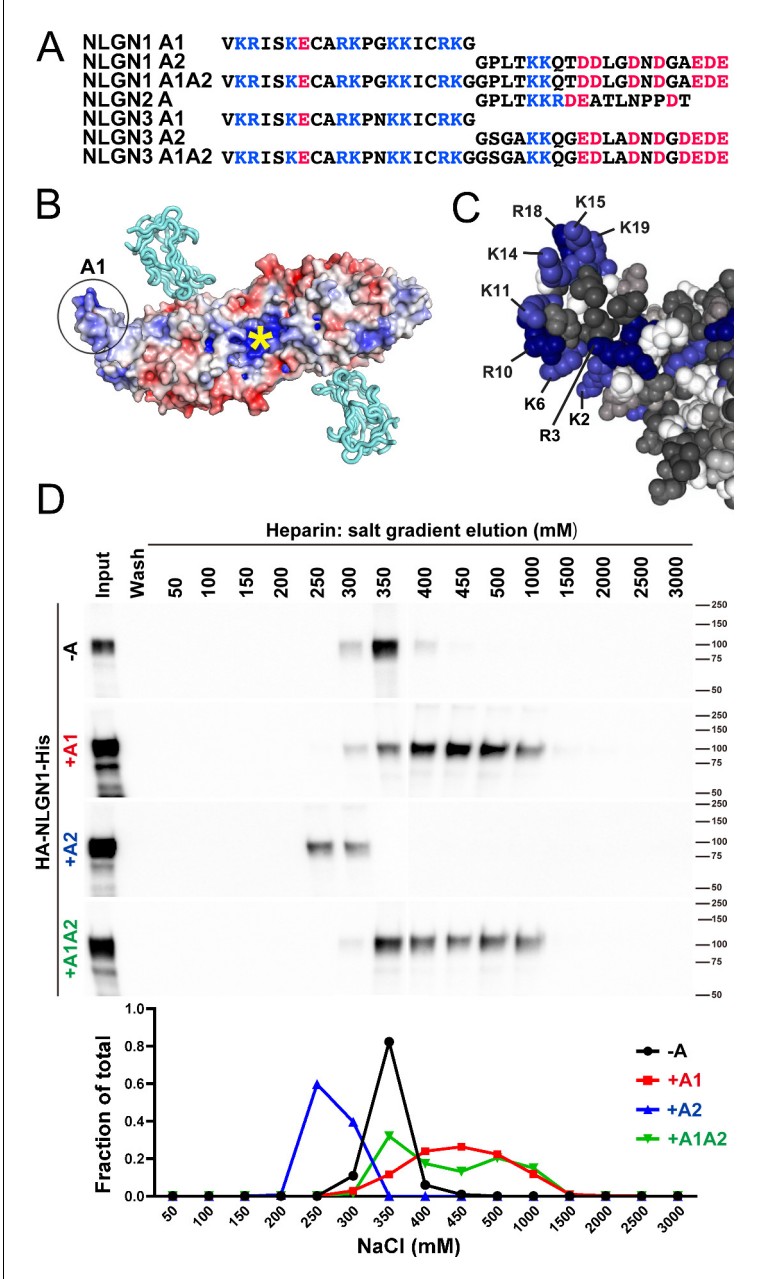

**Figure 1.** NLGN1 alternative splicing at site A regulates heparin/HS binding. (**A**) The amino acid sequences of each human NLGN A splice insert highlighting positively charged (blue) and negatively charged (red) residues. Splice insert sequences in mouse NLGN1-3 are identical to those in human NLGN1-3 except for one residue in NLGN1 A2 (7th residue H in mouse and Q in human). (**B**) Structure of the NLGN1-NRXN1β LNS domain complex (PDB: 3VKF) (*Tanaka et al., 2012*) showing the position of the A1 insert relative to the constitutive HS binding site (yellow asterisk) (*Zhang et al., 2018*). The NLGN1 surface is colored according to the electrostatic potential from blue (+8 kbT/ec) to red (−8 kbT/ec), and the NRXN LNS domain is in cyan. (**C**) Structure of the NLGN1 A1 splice insert (PDB: 3VKF) highlighting the positively charged surface residues proposed to participate in HS interaction. Residues are numbered from the beginning of the A1 insert. (**D**) Elution profile of purified recombinant NLGN1 isoform ectodomain proteins from a heparin column. Elution at a higher concentration of salt indicates stronger binding.

The online version of this article includes the following figure supplement(s) for figure 1:

**Figure supplement 1.** *NLGN* gene structures.

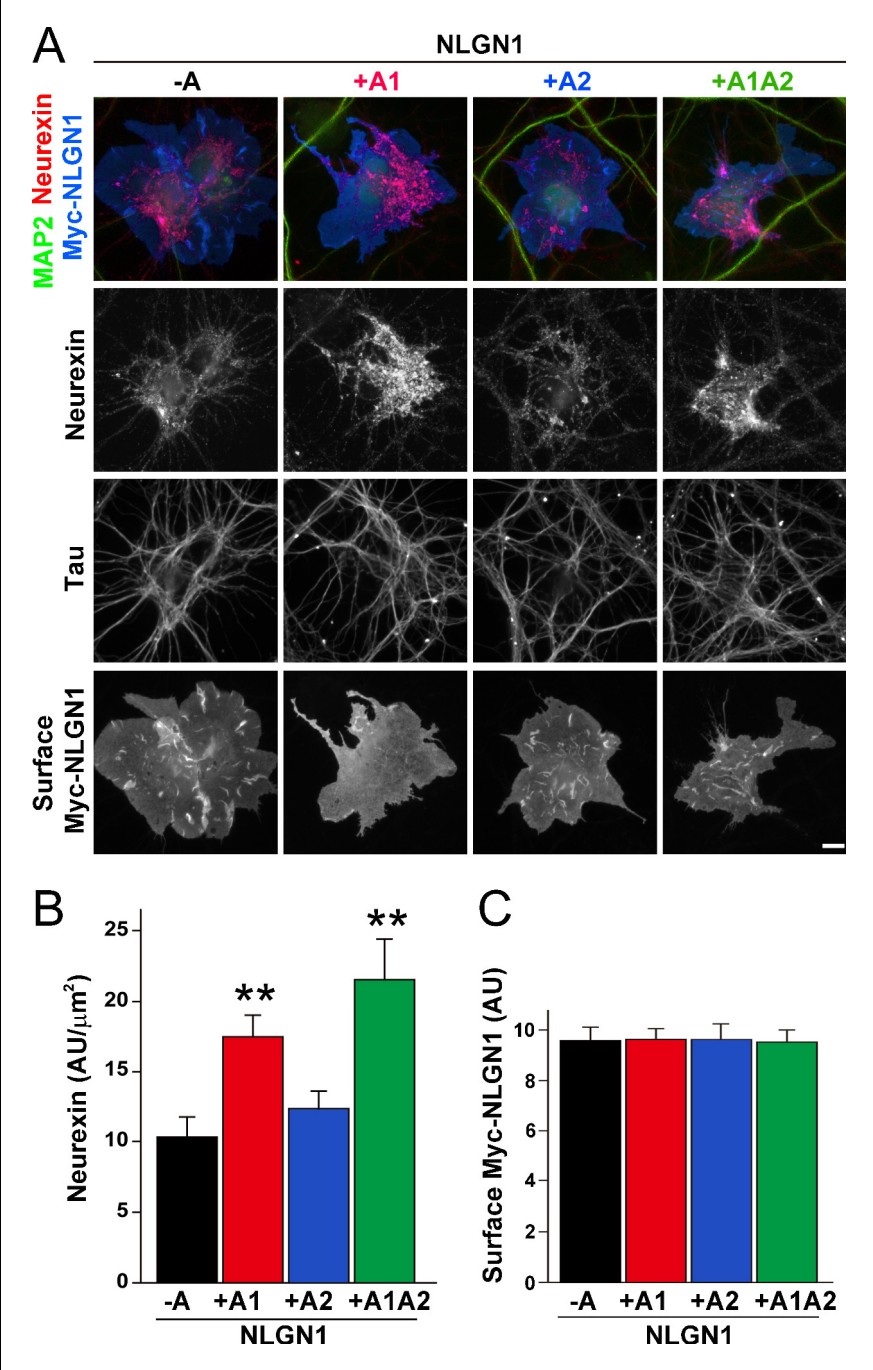

**Figure 2.** The NLGN1 A1 splice insert enhances neurexin recruitment in coculture. (**A–C**) Each NLGN1 isoform with an extracellular Myc tag was expressed in COS7 cells which were cocultured with hippocampal neurons. NLGN1 induced recruitment of neurexin along tau-positive axons. Axon regions contacting expressing COS7 cells and lacking contact with MAP2-positive dendrites were assessed to exclude native synapses. The total intensity of native neurexin (**B**) per contact area was normalized to a baseline value of 1 measured from sister cocultures performed with the negative control protein Myc-tagged Amigo. Neurexin recruitment differed among NLGN1 splice variants, p<0.0005 by Kruskal-Wallis test and **p<0.01 compared with NLGN1 -A by post hoc Dunn's multiple comparisons test, n = 35–43 cells from four independent experiments. Although the value for NLGN1 +A1A2 was higher than that for NLGN1 +A1, this difference was not significant. COS7 cells were chosen for equal surface NLGN1 expression (**C**). Scale bar, 10 µm.

The online version of this article includes the following source data for figure 2:

**Source data 1.** Source data for *Figure 2B,C*.

NRXN1β ectodomain containing the full LNS domain but lacking the HS modification site. One of these studies using bacterially expressed NRXN1β reported a positive effect of the A1 insert (*Comoletti et al., 2006*) although two studies using NRXN1β expressed in mammalian cells reported no difference in affinity between NLGN1 -A and NLGN1 +A1 (*Elegheert et al., 2017*; *Koehnke et al., 2010*). Altogether, these previous results and our data suggest that the NLGN1 A1 insert does not affect binding to the NRXN LNS domain but increases binding to the NRXN HS modification.

## NLGN1 A1 inclusion is high in GABAergic cell types

Our data above show that alternative splicing at the NLGN1 A site affects molecular interactions. Yet, although NLGN1 transcripts containing the A1, A2 and A1A2 inserts have been detected, cell type specific splicing patterns have not been well explored. Thus, we analysed three deep RNA-seq datasets for cell type splicing patterns of the NLGN1 A site. *Supplementary file 1* summarizes all RNA-seq datasets analysed in this study. Two datasets for adult mouse cortex from the Allen Institute for Brain Science covered 23,822 single-cell transcriptomes with >100,000 reads per cell detecting approximately 9500 genes per cell (*Tasic et al., 2018*). Cells were isolated by FACS or manual picking following layer-enriched dissections from many Cre driver lines crossed with reporters for access to select rare cell types, and the resulting 133 transcriptomic cell types were identified by clustering analysis (*Tasic et al., 2018*). We analysed the primary data (Gene Expression Omnibus (GEO) GSE115746) to assess cell-type splicing at NLGN1 splice site A. In mouse primary visual cortex, NLGN1 A1 inserts were most prevalent in GABAergic cell types, including the well-studied VIP and parvalbumin classes (*Figure 3A*). Oligodendrocytes also showed high A1 inclusion whereas A1 was essentially absent from NLGN1 in astrocytes and oligodendrocyte precursor cells. Cell type specific splicing patterns were generally similar in anterior lateral motor cortex (*Figure 3B*) except here the *Meis2*-expressing divergent GABAergic neuron class showed more NLGN1 A1 inclusion. We further analysed an independent dataset of ribosome-engaged transcripts from genetically defined neuron types in mouse cortex and hippocampus (*Furlanis et al., 2019*). Their deep RNA-seq designed to study splice isoforms covered >100 million reads per biological replicate, detecting >12,000 genes per sample with full-length coverage across transcripts. In this dataset of actively translated transcripts (GEO GSE133291), we again found that NLGN1 A1 inclusion was highest in GABAergic neurons of the cortex, including VIP and parvalbumin classes (*Figure 3C*).

To determine whether NLGN1 A1 inclusion changes with development, we assessed two deep RNA-seq datasets from mouse cortex spanning from embryonic day (E)14.5 to 2 years (*Lister et al., 2013*; *Yan et al., 2015*; *Supplementary file 1*). Previous analysis demonstrated dynamic splicing regulation of cassette exons in embryonic and postnatal cortex up to one month old (*Weyn-Vanhentenryck et al., 2018*). We estimated the abundance of NLGN1 isoforms by a regression analysis using reads mapped to exon junctions of the alternatively spliced region. Developmentally, NLGN1 A1 inclusion in mouse cortex peaked during the first to second postnatal weeks (*Figure 4A, B*).

## NLGN1 A1 inclusion is regulated by Rbfox

Cell-type specific alternative splicing is controlled by RNA-binding proteins that recognize specific regulatory sequences embedded in the pre-mRNA transcripts. Splicing factors specifically expressed or enriched in neurons include Rbfox, Mbnl, and Ptbp2 which have all been demonstrated to regulate alternative splicing of numerous neuronal transcripts (*Raj and Blencowe, 2015*; *Vuong et al., 2016*). Thus, we determined whether genetic knockout (KO) of these families of splicing factors altered splicing of NLGN1 A1.

The three Rbfox family members are thought to have considerable functional redundancy as they all bind to the same RNA sequence motif and have overlapping expression patterns in neurons (*Vuong et al., 2016*). To assess their combined role in NLGN1 A site splicing, due to limitations with lethality *in vivo*, we studied triple KO of *Rbfox1*, *Rbfox2* and *Rbfox3* in spinal neurons differentiated from mouse embryonic stem cells (*Jacko et al., 2018*; *Supplementary file 1*). At 5 days in culture, the embryonic stem cell-derived neurons elaborate branched processes and fire trains of action potentials upon current injection. Over the next 5 days of maturation, they show splicing changes paralleling those of early postnatal mouse cortex (*Jacko et al., 2018*). Our analyses of this dataset

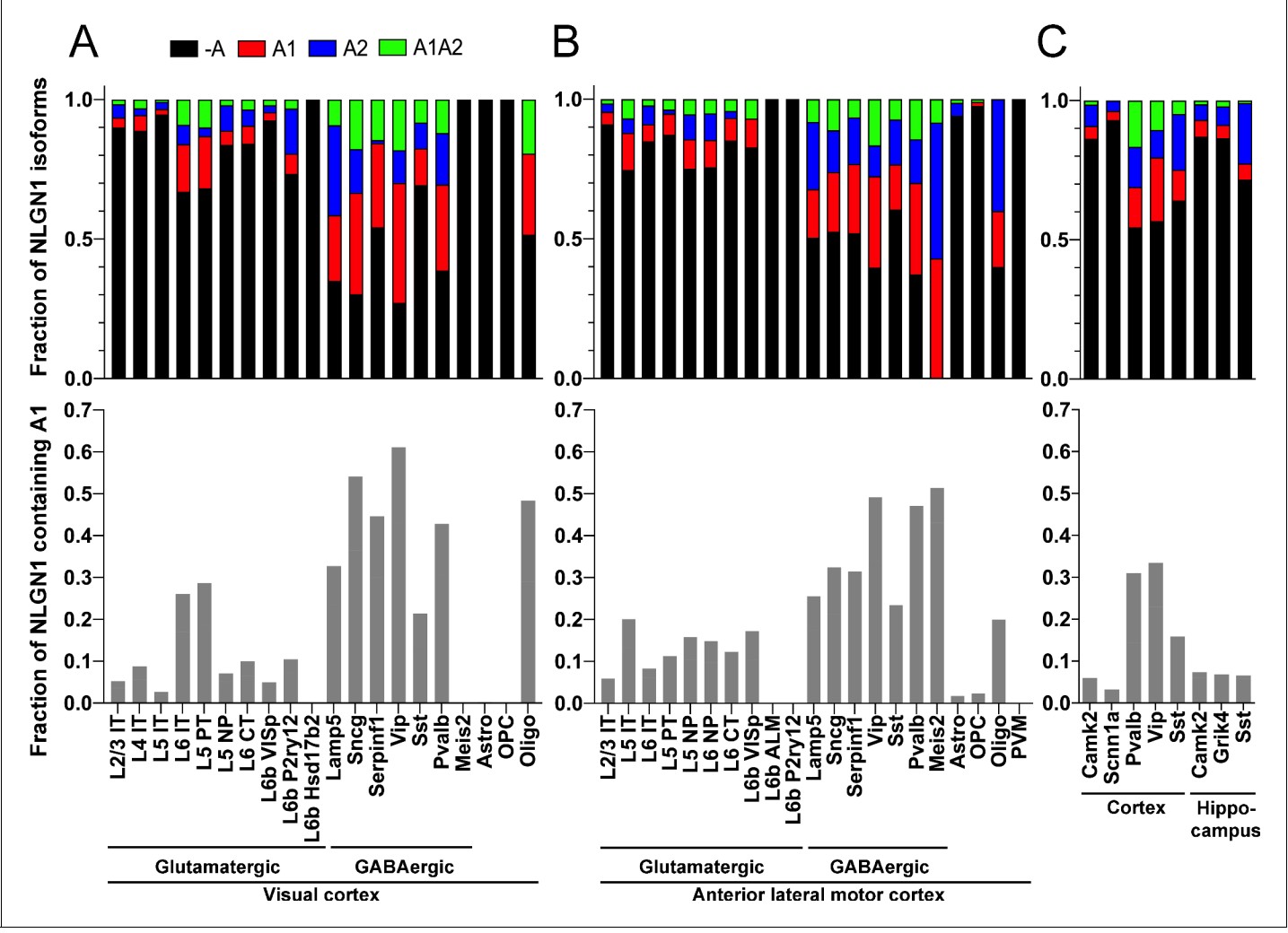

**Figure 3.** The NLGN1 A1 splice insert is high in GABAergic neuron cell types. The fraction of each mouse NLGN1 site A splice isoform transcript is plotted in each upper graph and the fraction of NLGN1 transcript that contains the A1 insert, that is (+A1 plus +A1A2)/total, is plotted in each lower graph. Datasets are from *Tasic et al., 2018* (A, B) and *Furlanis et al., 2019* (C).

revealed that *Rbfox* triple KO effectively eliminated NLGN1 A1 inclusion at both stages of neuron maturation (*Figure 4C*).

To identify additional RNA binding proteins which might regulate NLGN1 splicing, we similarly examined published RNA-seq data derived upon depletion of individual RNA binding proteins (*Supplementary file 1*). The data were derived from: adult hippocampus of *Mbnl2* KO mice (*Charizanis et al., 2012*); adult frontal cortex of *Mbnl1*$^{-/-}$ *Mbnl2*$^{loxP/loxP}$ Nestin-Cre nervous system-specific double KO mice (*Weyn-Vanhentenryck et al., 2018*); embryonic day 18 brains of *Ptbp2*$^{loxP/loxP}$ Nestin-Cre nervous system-specific KO mice, and postnatal day 1 cortex of *Ptbp2*$^{loxP/loxP}$ Emx1-Cre forebrain glutamatergic neuron-selective KO mice (*Li et al., 2014*). These datasets were analysed for alternative splicing in our previous study (*Weyn-Vanhentenryck et al., 2018*), from which we obtained reads mapped to the alternatively spliced region of NLGN1 to estimate abundance of different splice variants. Our analyses of these datasets essentially rules out a role for Mbnl in NLGN1 A site splicing (*Figure 4D*). For Ptbp2, the Emx1-Cre KO datasets did not show any difference in NLGN1 A1 splicing, but NLGN1 A1 inclusion was reduced in the Nestin-Cre KO dataset (*Figure 4E*). These findings suggest Ptbp2 may regulate NLGN1 -A1 splicing in neurons outside the forebrain. Therefore, our transcriptomic analyses indicated a major role for Rbfox and lesser roles, if any, for Mbnl or Ptbp2 in NLGN1 A1 splicing.

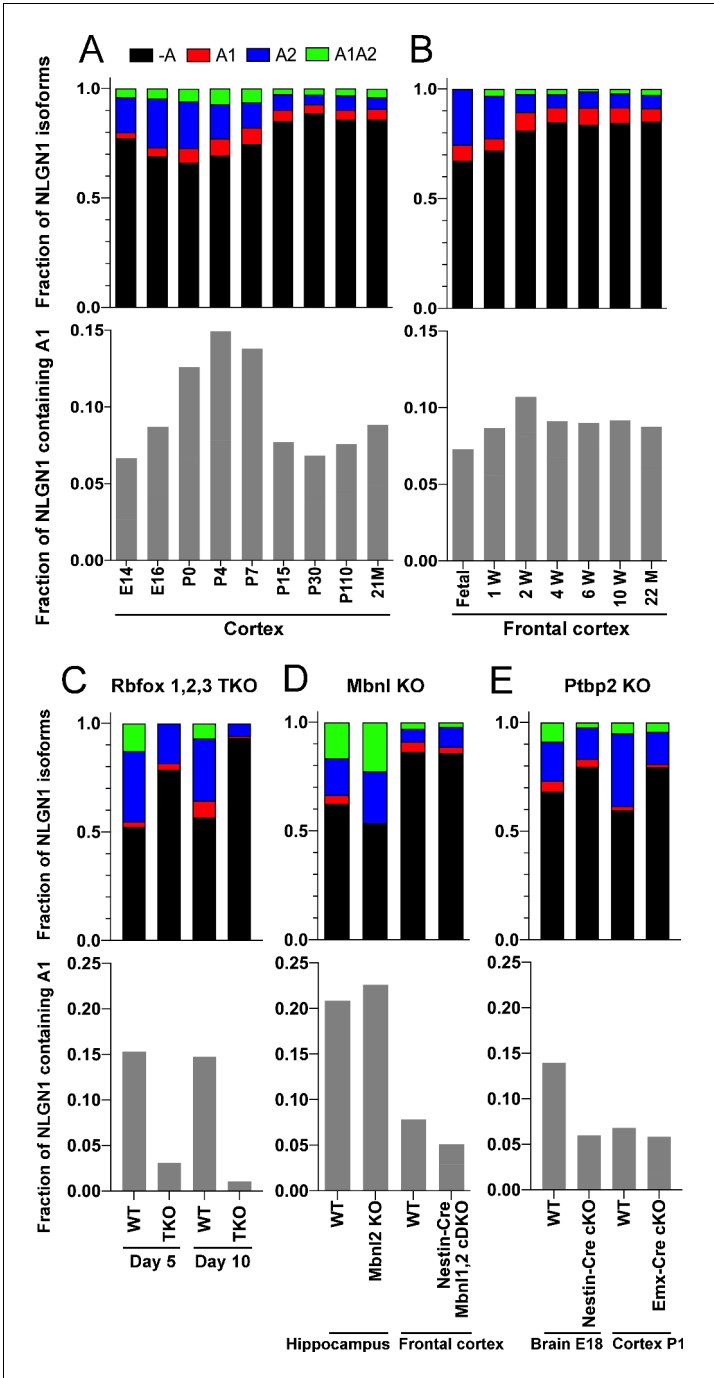

**Figure 4.** The NLGN1 A1 splice insert is regulated developmentally and by Rbfox splicing factors. The fraction of each mouse NLGN1 site A splice isoform transcript is plotted in each upper graph and the fraction of NLGN1 transcript that contains the A1 insert, that is (+A1 plus +A1A2)/total, is plotted in each lower graph. In the developmental studies (**A, B**), ages are indicated in embryonic (**E**) or postnatal (**P**) days, or in postnatal weeks (**W**) or months (**M**). Panel (**C**) data are from spinal neuron cultures differentiated from *Rbfox1,2,3* triple KO (TKO) embryonic stem cells and grown for the indicated number of days. Panel (**D**) data are from hippocampi from 2 to 3 month old *Mbnl2* KO mice or frontal cortex from adult *Mbnl1−/− Mbnl2loxP/loxP* Nestin-Cre conditional double KO (cDKO) mice. Panel (**E**) data are from embryonic day 18 brain of *Ptbp2loxP/loxP* Nestin-Cre cKO mice or postnatal day 1 cortex of *Ptbp2loxP/loxP* Emx1-Cre cKO mice. Development datasets are from (**A**) (*Yan et al., 2015*) and (**B**) (*Lister et al., 2013*). Datasets from KO cells or mice lacking splice factors are from (**C**) (*Jacko et al., 2018*), (**D**) (*Charizanis et al., 2012*; *Weyn-Vanhentenryck et al., 2018*) and (**E**) (*Li et al., 2014*).

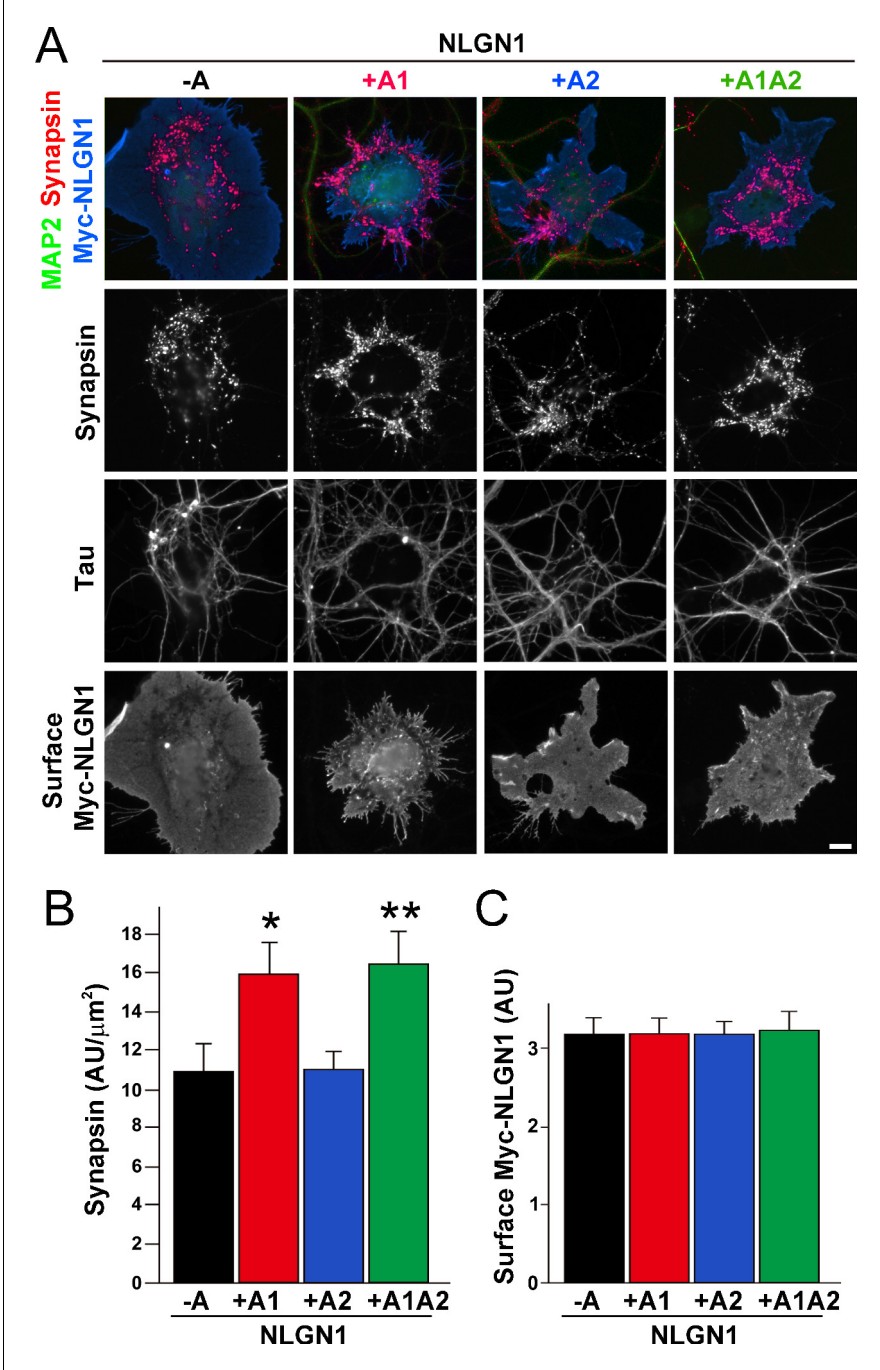

**Figure 5.** The NLGN1 A1 splice insert enhances presynaptic differentiation in coculture. (**A–C**) Each NLGN1 isoform with an extracellular Myc tag was expressed in COS7 cells which were cocultured with hippocampal neurons. NLGN1 induced clustering of synapsin along tau-positive axons. Axon regions contacting expressing COS7 cells and lacking contact with MAP2-positive dendrites were assessed to exclude native synapses. The total intensity of synapsin (**B**) per contact area was normalized to a baseline value of 1 measured from sister cocultures performed with the negative control protein Myc-tagged Amigo. Synapsin clustering differed among NLGN1 splice variants, p<0.001 by Kruskal-Wallis test and *p<0.05 and **p<0.01 compared with NLGN1 -A by post hoc Dunn's multiple comparisons test, n = 47–56 cells from four independent experiments. COS7 cells were chosen for equal surface Myc-NLGN1 expression (**C**). Scale bar, 10 μm.

The online version of this article includes the following source data for figure 5:

**Source data 1.** Source data for *Figure 5B,C*.

## The NLGN1 A1 insert enhances presynaptic differentiation in coculture

To test the functional impact of the NLGN1 A1 splice site, we first used the neuron - COS7 cell coculture assay to assess synaptogenic activity of the NLGN1 splice variants. Via NRXN recruitment as seen in *Figure 2*, NLGNs expressed on the surface of non-neuronal cells induce full presynaptic differentiation in contacting axons, reflected by clustering of multiple presynaptic components (*Scheiffele et al., 2000*). NLGN binding to axonal NRXN is required for this synaptogenic activity in coculture, based on loss of activity upon NRXN triple knockdown (*Gokce and Südhof, 2013*; *Zhang et al., 2018*) or by NLGN point mutations that disrupt binding to the NRXN LNS domain (*Ko et al., 2009*) or to the NRXN HS modification (*Zhang et al., 2018*). Here we found that NLGN1 +A1 and +A1A2 induced greater presynaptic differentiation than other splice variants, assessed by synapsin clustering at axon - COS7 cell contact sites lacking dendrite contact to exclude native synapses (*Figure 5*). Thus, the NLGN1 -A1 insert promotes presynaptic differentiation in coculture assays.

## The NLGN1 A1 insert promotes structural and functional synapse development

To assess the role of the NLGN1 A1 insert at native synapses, we used a molecular replacement strategy knocking down native NLGN1 with sh*Nlgn1* (*Zhang et al., 2018*) and re-expressing RNAi-resistant Myc-NLGN1* lacking or containing the A1 insert. For these assays, we focused on NLGN1 -A, the major neuronal form (*Figures 3* and *4*; *Chih et al., 2006*), and NLGN1 +A1. We found that NLGN1* +A1 induced greater presynaptic differentiation than NLGN1* -A, assessed by total intensity of vesicular glutamate transport VGlut1 clustering onto expressing neurons (*Figure 6A–D*). There was no difference between the NLGN1 isoforms in the density of VGlut1 clusters per dendrite area, suggesting no difference in synapse numbers, but a difference in total intensity of VGlut1 per cluster.

We observed no difference in the localization of Myc-NLGN1* -A and +A1 in these assays. However, as in previous assays expressing YFP-NLGN1 in the absence of recombinant postsynaptic scaffolds (*Graf et al., 2004*), both isoforms of Myc-NLGN1* showed high diffuse levels with poor postsynaptic clustering. Thus, to determine whether A1 alternative splicing mediates differential synaptic recruitment, we used a more sensitive tag on NLGN1* and slightly different culture conditions (*Chamma et al., 2016*; *Letellier et al., 2018*). NLGN1* was tagged with an extracellular 15-amino-acid acceptor peptide (AP) that is biotinylated upon co-expression of a biotinylating enzyme BirA[ER] and then surface labelled with fluorescent streptavidin. The longer culture time and co-expression of Homer1c-dsRed to mark postsynaptic sites may also enhance synapse maturation (*Zeng et al., 2018*), altogether resulting in postsynaptic clustering of AP-NLGN1*. However, there was no difference in the degree of synaptic enrichment of AP-NLGN1* -A versus +A1, nor in Homer1c-dsRed synaptic enrichment, cluster density, or cluster area between groups (*Figure 6E–I*). Collectively, our data indicate that the NLGN1 A1 splice insert mediates differential recruitment of NRXN and presynaptic differentiation but not differential postsynaptic targeting.

To assess effects of NLGN1 A1 splicing directly on synaptic function, we used a similar molecular replacement strategy to knockdown native NLGN1 and re-express NLGN1* -A or +A1 and recorded miniature excitatory synaptic currents (mEPSCs) in cultured hippocampal neurons. Neurons expressing sh*Nlgn1* and rescued with NLGN1* +A1 had significantly greater mEPSC frequency than those rescued with NLGN1* -A, with a corresponding reduction in mEPSC interevent interval (*Figure 7*). There was no difference between groups in mEPSC amplitude. Thus, the NLGN1 A1 splice insert enhances functional transmission as well as structural presynaptic differentiation.

## A1 insertion partially restores autism-linked NLGN1 P89L synaptogenic function

We propose that the A1 insert might be a potential target for alleviating deficits of ASD *NLGN1* missense mutations by enhancing their function. To test this idea, we chose the *NLGN1* P89L missense variant found in two ASD siblings but not in controls (*Nakanishi et al., 2017*). Modelling in *Nlgn1* P89L heterozygous knock-in mice resulted in reduced NLGN1 protein levels and deficits in social interaction, altered social dominance, and impaired spatial memory, supporting a causative role of this mutation in autism (*Nakanishi et al., 2017*). We confirmed an expected reduction in surface

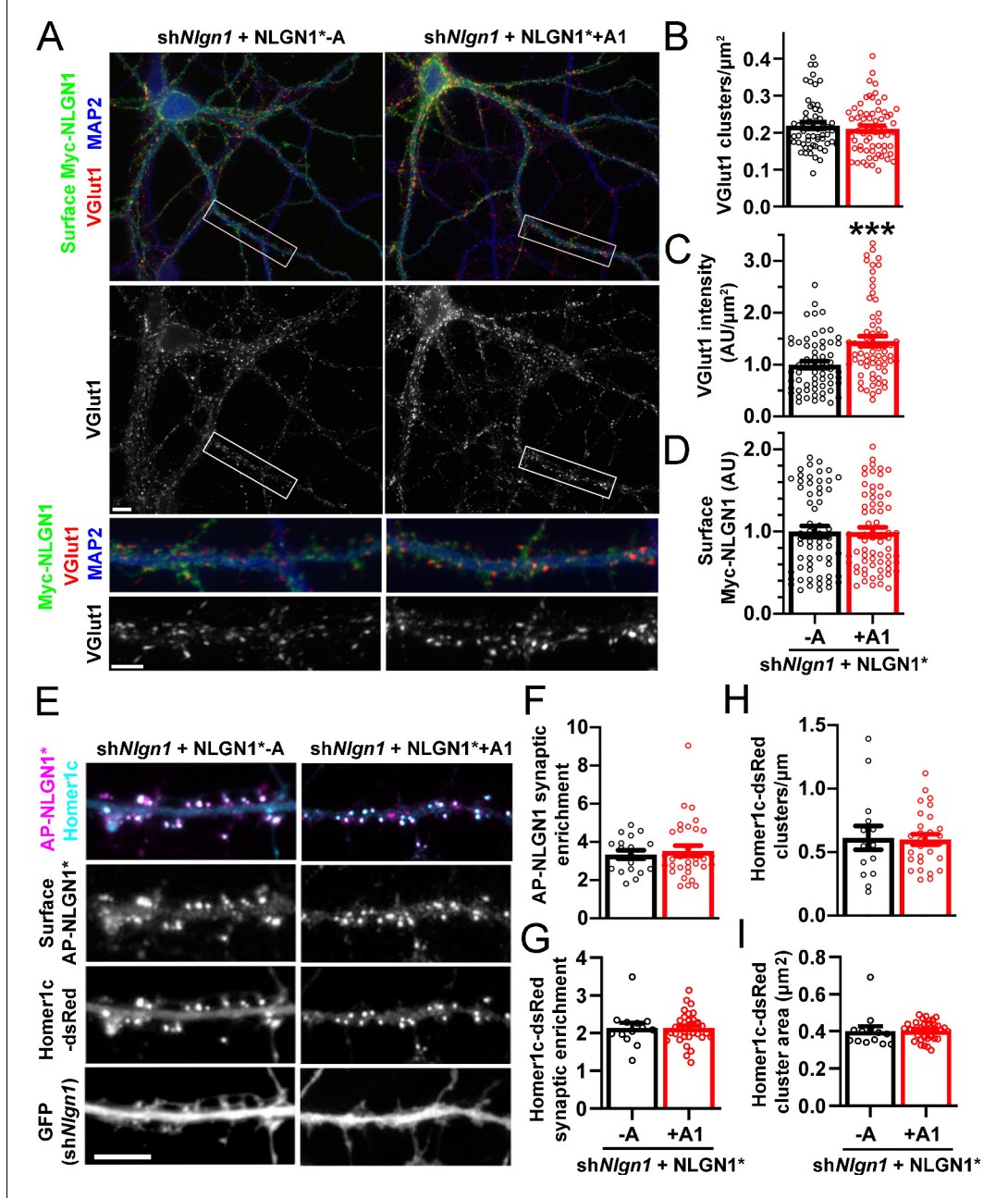

**Figure 6.** The NLGN1 A1 splice insert promotes structural synapse development. (A–D) Cultured hippocampal neurons were transfected with U6-sh*Nlgn1*-hSyn-CFP and hSyn-Myc-NLGN1* -A or +A1 at DIV 5 and neurons were fixed at DIV 12. The density of VGlut1 clusters did not differ but the total fluorescence intensity of VGlut1 inputs was higher for neurons expressing the NLGN1* +A1 than the -A isoform, ***p=0.0006 by Mann Whitney test, n = 59–66 neurons from three independent experiments. Cells were chosen for equal intensity of surface Myc-NLGN1. Scale bar, top 10 μm, bottom 5 μm. (E–I) Cultured hippocampal neurons were electroporated at plating with sh*Nlgn1*-GFP, Homer1c-dsRed, BirA[ER], and hSyn-AP-NLGN1* -A or +A1 and neurons imaged at DIV 14 following live cell labeling with streptavidin-Alexa647. The synaptic enrichment of AP-NLGN1*, defined as the intensity of AP-NLGN1* in Homer1c-dsRed-positive clusters relative to the intensity in the local dendrite shaft, did not differ between the splice variants, nor was there any difference in the synaptic enrichment or density or mean area of Homer1c-dsRed clusters (all p>0.1 by Mann Whitney test, n > 14 neurons from two independent experiments). Scale bar, 10 μm.

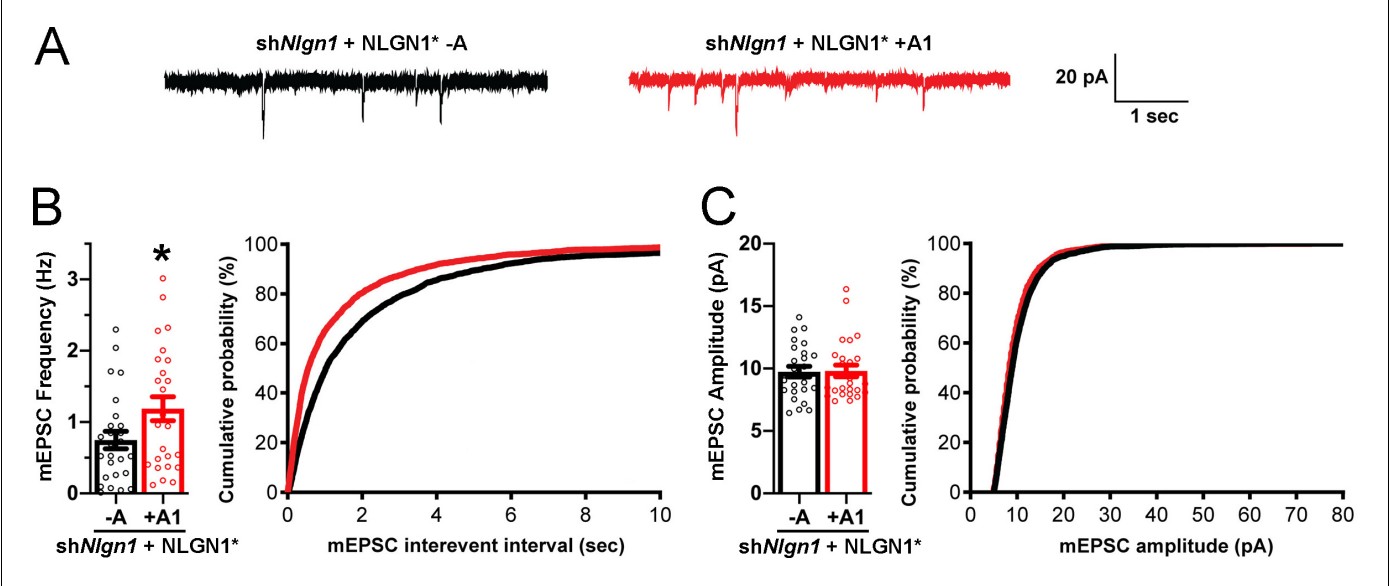

**Figure 7.** The NLGN1 A1 splice insert promotes functional synapse development. Cultured hippocampal neurons were transfected with U6-sh*Nlgn1*-hSyn-YFP, hSyn-DIO-YFP-P2A-HA-NLGN1* -A or +A1, and CAG-Cre at DIV 5. YFP positive neurons were selected for mEPSC recording at DIV 13 and 14. (**A**) Representative mEPSC traces from neurons expressing NLGN1* lacking (black trace) or containing (red trace) the A1 splice variant. (**B**) mEPSC frequency was significantly increased in NLGN1* +A1-expressing neurons (n = 26) relative to cells expressing NLGN1* -A (n = 26; *p=0.040 by Welch's t test), with a corresponding change in interevent interval. (**C**) mEPSC amplitude did not significantly differ between groups (p=0.93). Scale bar, 20 pA, 1 s.

expression of Myc-NLGN1 -A P89L relative to WT (*Figure 8—figure supplement 1*), likely reflecting enhanced endoplasmic reticulum-associated degradation (*Nakanishi et al., 2017*).

We next assessed the synaptogenic activity of NLGN1 P89L. Interestingly, even though cells with equal NLGN1 surface expression were chosen for the coculture assay, the P89L mutation significantly reduced presynaptic differentiation by NLGN1 (*Figure 8*). NLGN1 P89L -A induced <20% of the level of synapsin clustering as wild type NLGN1 -A, even after controlling for any difference in surface expression. Thus, the P89L mutation not only reduces total and surface levels of NLGN1 (*Nakanishi et al., 2017*; *Figure 8—figure supplement 1*) but also renders the NLGN1 that reaches the cell surface less effective at inducing presynaptic differentiation.

Importantly, addition of the A1 insert partially alleviated this deficit: NLGN1 P89L +A1 induced greater presynaptic differentiation than NLGN1 P89L -A (*Figure 8*). Furthermore, in the native synapse molecular replacement assay with NLGN1 knockdown, the total fluorescence intensity of VGlut1 inputs was higher onto hippocampal neurons expressing NLGN1* P89L +A1 than NLGN1* P89L -A (*Figure 9*). Thus, the A1 insert functions in the context of the P89L ASD mutation to enhance presynaptic differentiation.

## Discussion

In contrast to the well-studied NLGN1 B splice site, the A splice site has been poorly studied, with information previously lacking on the A1 insert. Yet the A1 insert appears highly conserved among vertebrates, including humans, mice, zebrafish, and chickens (*Figure 1A*; *Rissone et al., 2010*; *Wahlin et al., 2010*), suggesting a functional importance. We show here that the A1 insert increases binding of NLGN1 to heparin and increases recruitment of the axonal binding partner of NLGN1, HS-modified NRXN. It will be important in future studies to directly compare the affinities of purified NLGN1 splice forms with native HS-modified NRXN, such as with a neuron-based binding assay. Our data reveal an intersection between two forms of post-transcriptional modification, alternative splicing of NLGN1 and glycan modification of NRXN. Functionally, NLGN1 A1 inclusion increases presynaptic differentiation and synaptic transmission but does not affect NLGN1 synaptic targeting or synapse density. Data from KO neurons reveals regulation of A1 splicing by the Rbfox family of

splicing factors, and transcriptomic analyses reveals high A1 inclusion in multiple cortical GABAergic cell types. Finally, we show that A1 inclusion enhances function not just of wild type NLGN1 but also of the ASD-linked NLGN1 P89L mutant. We suggest that the A splice site might be harnessed as a means to restore function in disorders involving deleterious mutations in *NLGN1,* and potentially *NLGN3,* by developing therapeutics to convert the more prevalent -A form to the more synaptogenic +A1 form.

The role identified here for the NLGN1 splice site A1 insert is an unusual one, increasing NLGN1 interaction with HS glycan and thus enhancing synaptogenic function. Although there are many cases of alternative splicing regulating synapse development through altered protein-protein interactions, a mechanism involving altered protein-glycan interaction is less common. The NLGN1 B splice site also regulates synaptogenic function partially through glycosylation, in that case through N-glycosylation of the NLGN1 B insert blocking binding to α-NRXN while retaining binding to β-NRXN (*Boucard et al., 2005*). Neural-specific alternative splicing of agrin at its y site also regulates its

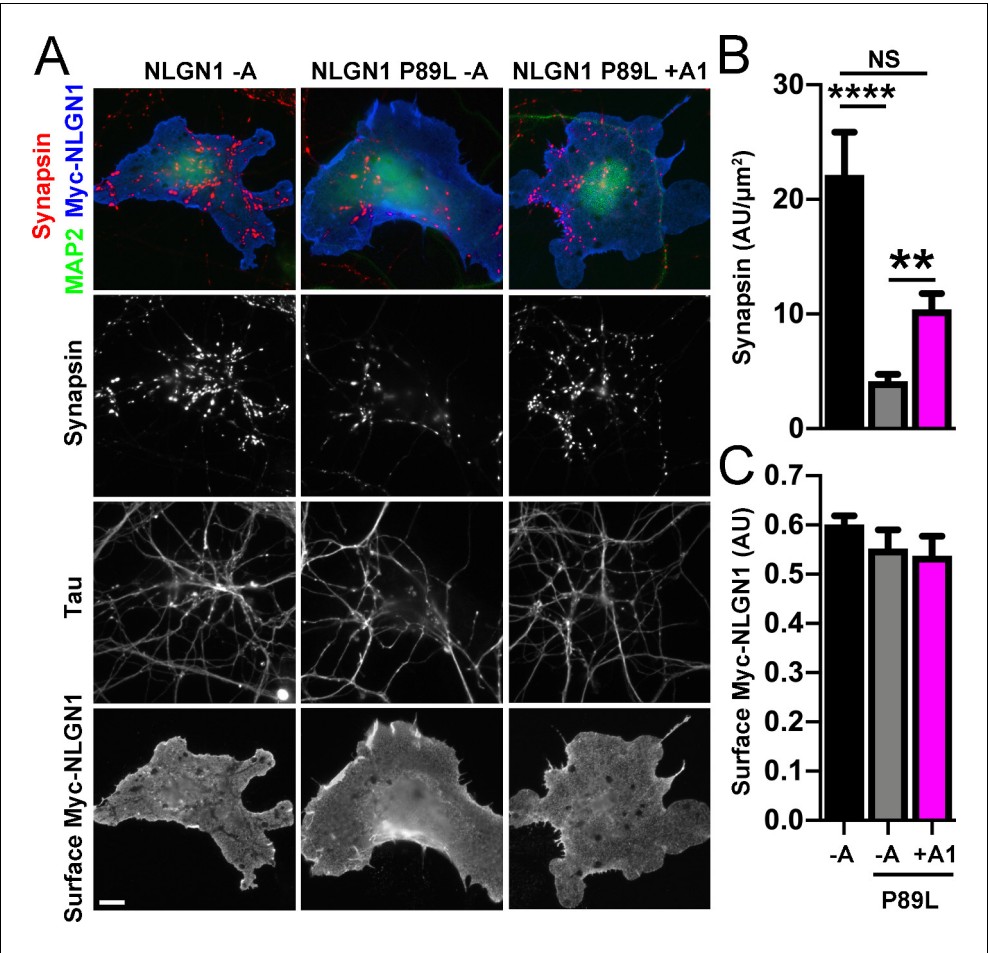

**Figure 8.** A1 splice inclusion in the autism-linked NLGN1 P89L mutant enhances presynaptic differentiation in coculture. (**A–C**) Myc-NLGN1 -A, Myc-NLGN1 P89L -A, or Myc-NLGN1 P89L +A1 was expressed in COS7 cells which were cocultured with hippocampal neurons. The total intensity of synapsin (**B**) per COS7 cell - axon contact area lacking contact with MAP2-positive dendrites was normalized to a baseline value of 1 measured from sister cocultures performed with the negative control protein Myc-tagged CD4. Synapsin clustering by NLGN1 -A was reduced by the P89L mutation and partially restored by A1 splice site inclusion, ****p<0.0001, **p<0.01, NS not significant by Kruskal-Wallis test with post hoc Dunn's multiple comparisons test. n = 18–29 cells from two independent experiments. COS7 cells were chosen for equal NLGN1 expression (**C**). Scale bar, 10 μm.

The online version of this article includes the following source data and figure supplement(s) for figure 8:

**Source data 1.** Source data for *Figure 8B,C*.
**Figure supplement 1.** The NLGN1 P89L mutation reduces surface expression.

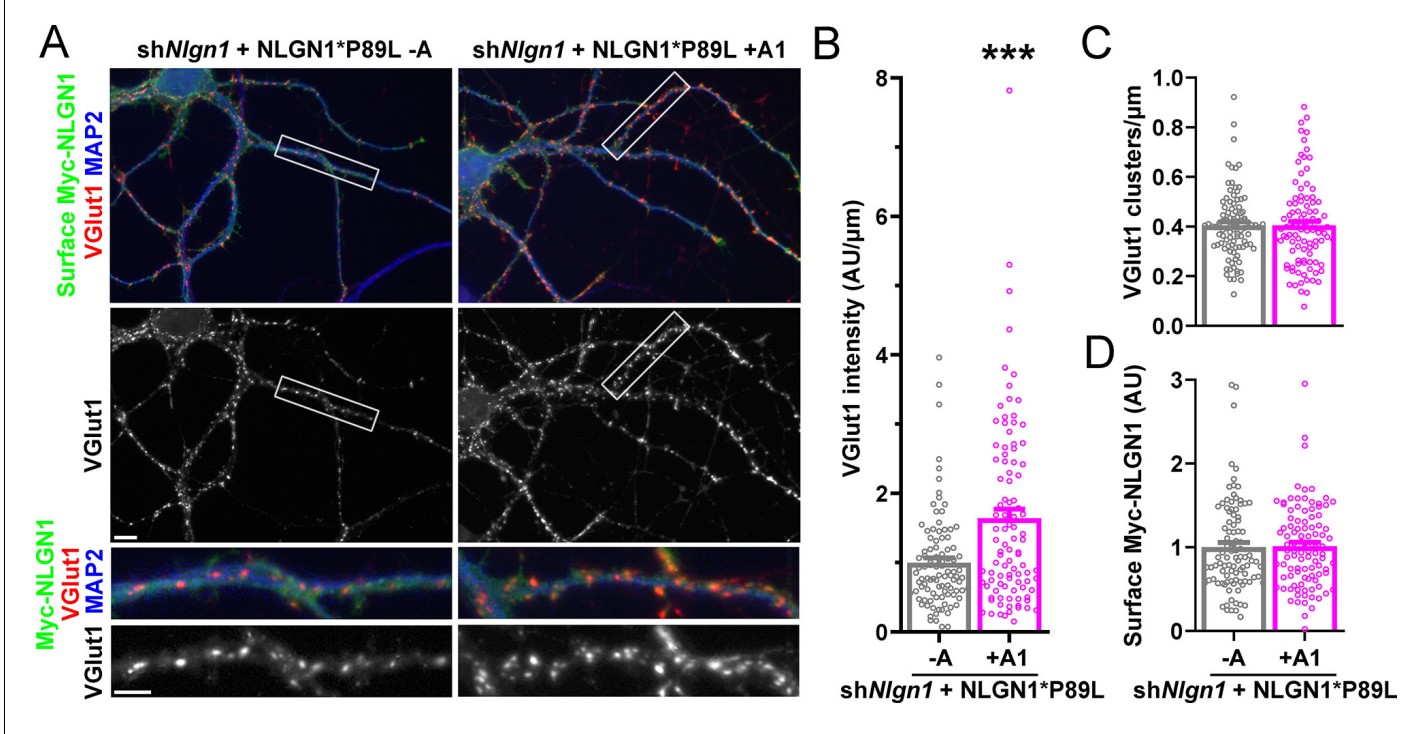

**Figure 9.** A1 splice inclusion in the autism-linked NLGN1 P89L mutant promotes synapse development. (**A–D**) Cultured hippocampal neurons were transfected with U6-sh*Nlgn1*-hSyn-YFP and hSyn-CFP-P2A-Myc-NLGN1* P89L -A or P89L +A1 at DIV 5 and neurons were fixed at DIV 13. The density of VGlut1 clusters did not differ (**C**) but the total fluorescence intensity of VGlut1 inputs (**B**) was higher for neurons expressing NLGN1* P89L +A1 than the -A isoform, ***p=0.0008 by Mann Whitney test, n = 97–99 neurons from three independent experiments. Cells were chosen by equal surface Myc-NLGN1 immunofluorescence (**D**). Scale bar, top 10 µm, bottom 5 µm.

interaction with HS, and HS binding reduces agrin interaction with dystroglycan (*Gesemann et al., 1996*; *O'Toole et al., 1996*). However, the y site and these interactions are not critical for agrin synaptogenic activity (*Burgess et al., 1999*; *Li et al., 2018*), unlike the central role of the NRXN interaction for NLGN1 function.

Our data reveal effects of the NLGN1 A1 insert on increasing NRXN recruitment and presynaptic differentiation in coculture, and on increasing VGlut1 levels at inputs and mEPSC frequency in hippocampal neurons. However, the NLGN1 A1 insert had no apparent effect on the synaptic targeting of NLGN1. This finding is consistent with other data suggesting that the excitatory postsynaptic localization of NLGN1 is largely controlled by intracellular interactions regulated by multiple kinases (*Jeong et al., 2019*; *Jeong et al., 2017*; *Letellier et al., 2020*; *Letellier et al., 2018*). The NLGN1 +A1-associated increase in mEPSC frequency and VGlut1 cluster intensity without a change in VGlut1 or Homer1c-dsRed cluster density suggest there is no change in synapse numbers but rather an increase in neurotransmitter release. These findings are consistent with previous reports of a retrograde effect of NLGN1 on neurotransmitter release probability (*Futai et al., 2007*; *Stan et al., 2010*). In contrast, the lack of effect of the NLGN1 A1 insert on mEPSC amplitude suggests no effect on postsynaptic sensitivity. Thus, altogether our data are consistent with a purely retrograde mechanism of action of the NLGN1 A1 insert through enhanced NRXN recruitment mediating enhanced presynaptic differentiation and increased transmitter release rate. While this work was under revision, it was reported that the A1 insert in NLGN3 regulates GABAergic transmission, as overexpression of NLGN3 +A1 or +A1A2 reduced GABAergic transmission whereas NLGN3 -A or +A2 increased GABAergic transmission relative to neighbour untransfected cells in hippocampal slice cultures (*Uchigashima et al., 2020*). Differential effects were also found on presynaptic release probability. We suggest that the mechanism of this effect may also involve differential interactions with NRXN -HS.

The effect of the higher NLGN1 A1 inclusion in GABAergic than in glutamatergic neuron classes may be to increase excitatory input selectively onto the GABAergic neurons, thus enhancing synchrony of the overall network. In particular, cortical VIP and parvalbumin interneurons, which showed high NLGN1 A1 inclusion in all datasets, play key roles in neural synchrony and cognitive processing. For example, VIP interneuron disinhibition of prefrontal responses to hippocampal inputs controls avoidance behaviour (*Lee et al., 2019*). NLGN3 on parvalbumin interneurons contributes to gamma oscillations and social behaviour as well as fear conditioning (*Cao et al., 2018*; *Polepalli et al., 2017*).

Our data also shed some light on the mechanisms regulating alternative splicing of NLGN1 A1, with KO data supporting a role for Rbfox and a possible lesser role for Ptbp2 splice factors. Although further validation of NLGN1 as an Rbfox target will be needed, NLGN1 appears to be one among many Rbfox targets. Unlike the highly dedicated splicing program of SLM2 regulating synaptic properties through NRXN site 4 splicing (*Traunmüller et al., 2016*), the Rbfox family regulates a broader range of targets controlling synaptic properties and other aspects of neuronal maturation including axon initial segment assembly (*Jacko et al., 2018*). Rbfox proteins also act as more than just splicing factors; in GABAergic neurons, a mechanism involving Rbfox1 binding to the Vamp1 mRNA 3' UTR regulates synaptic transmission (*Vuong et al., 2018*). Like *NLGN*s, *RBFOX1* is a risk gene for ASD based on strong evidence including *de novo* copy number deletions (*Sebat et al., 2007*; *Zhao, 2013*).

The results here with the P89L NLGN1 mutant support the idea that elevating A1 insertion is a promising direction to enhance the function of *NLGN1* in cases of deleterious mutations. This approach will likely extend to *NLGN3* as well as *NLGN1*, given their high sequence homology including the A1 insert (*Figure 1A*). *NLGN3* is among the top risk genes for ASD with multiple functionally deleterious missense mutations (*Jamain et al., 2003*; *Quartier et al., 2019*). In the majority of cases which involve heterozygous mutations, reagents that increase A1 insertion could act by enhancing function of both the mutant NLGN and the wild type allele. *NRXN1* is also among the top risk genes for ASD as well as for schizophrenia and Tourette's Syndrome (*Tourette Syndrome Association International Consortium for Genetics (TSAICG) et al., 2017*; *Rees et al., 2014*; *Südhof, 2017*). Therapies targeting wild type *NLGN* splicing to elevate the function of NLGN-NRXN complexes may also be partially efficacious in cases of *NRXN* mutations. Clinically effective oligonucleotide reagents targeting splicing events have reached the bedside for other disorders, notably for increasing specific exon inclusion in *SMN2* to treat spinal muscular atrophy (*Bennett et al., 2019*; *Singh et al., 2017*).

# Materials and methods

## Key resources table

| Reagent type (species) or resource | Designation | Source or reference | Identifiers | Additional information |
|---|---|---|---|---|
| Strain, strain background (*Rattus norvegicus*) | Sprague-Dawley | Charles River | CD-IGS | For primary hippocampal neuron cultures |
| Strain, strain background (*Mus musculus*) | C57BL/6J | Jackson Laboratory | Cat# 00064 | For RNA for RT-PCR for cloning |
| Cell line (*Homo sapiens*) | HEK293 | ATCC | Cat# CRL-1573 | |
| Cell line (*Cercopithecus aethiops*) | COS7 | ATCC | Cat# CRL-1651 | |
| Biological sample (*Rattus norvegicus*) | Primary hippocampal neurons | This paper | | Freshly isolated from *Rattus norvegicus* |
| Antibody | anti-Myc (rabbit polyclonal) | Sigma | Cat# C3956; RRID:AB_439680 | (ICC 1:40000) |
| Antibody | anti-neurexin (rabbit polyclonal) | Millipore | Cat# ABN161; RRID:AB_10917110 | (ICC 1:1000) |

*Continued on next page*

*Continued*

| Reagent type (species) or resource | Designation | Source or reference | Identifiers | Additional information |
|---|---|---|---|---|
| Antibody | anti-MAP2 (chicken polyclonal) | Abcam | Cat# ab5392; RRID:AB_2138153 | (ICC 1:80000) |
| Antibody | anti-VGlut1 (mouse monoclonal) | NeuroMab | Cat# N28/9; RRID:AB_2187693 | (ICC 1:4000) |
| Antibody | anti-synapsin I (mouse monoclonal) | Synaptic Systems | Cat# 106 011; RRID:AB_2619772 | (ICC 1:20000) |
| Antibody | anti-tau (mouse monoclonal) | Millipore | Cat# MAB3420; RRID:AB_94855 | (ICC 1: 2000) |
| Antibody | anti-HA (rat monoclonal) | Roche | Cat# 1186743100; RRID:AB_390919 | (WB 1: 2000) |
| Recombinant DNA reagent | pNICE-HA-NLGN1 | Scheiffele et al., 2000 | | To subclone NLGN1 isoforms |
| Recombinant DNA reagent | pCAGGS-Myc-NLGN1 (-A+B)WT | This paper | | To express Myc-NLGN1 -A |
| Recombinant DNA reagent | pCAGGS-Myc-NLGN1 (+A1+B)WT | This paper | | To express Myc-NLGN1 +A1 |
| Recombinant DNA reagent | pCAGGS-Myc-NLGN1 (+A2+B)WT | This paper | | To express Myc-NLGN1 +A2 |
| Recombinant DNA reagent | pCAGGS-Myc-NLGN1 (+A1A2+B)WT | This paper | | To express Myc-NLGN1 +A1A2 |
| Recombinant DNA reagent | pCAGGS-Myc-NLGN1 (-A+B)P89L | This paper | | To express Myc-NLGN1 P89L -A |
| Recombinant DNA reagent | pCAGGS-CFP-P2A-Myc-NLGN1(-A+B)WT | This paper | | To express Myc-NLGN1 -A |
| Recombinant DNA reagent | pCAGGS-CFP-P2A-Myc-NLGN1(-A+B)P89L | This paper | | To express Myc-NLGN1 P89L -A |
| Recombinant DNA reagent | pCAGGS-CFP-P2A-Myc-NLGN1(+A1+B)P89L | This paper | | To express Myc-NLGN1 P89L +A1 |
| Recombinant DNA reagent | pcDNA4-HA-ecto-NLGN1(-A+B)-His | This paper | | To express HA-NLGN1-His -A ectodomain |
| Recombinant DNA reagent | pcDNA4-HA-ecto-NLGN1(+A1+B)-His | This paper | | To express HA-NLGN1-His +A1 ectodomain |
| Recombinant DNA reagent | pcDNA4-HA-ecto-NLGN1(+A2+B)-His | Zhang et al., 2018 | | To express HA-NLGN1-His +A2 ectodomain |
| Recombinant DNA reagent | pcDNA4-HA-ecto-NLGN1(+A1A2+B)-His | This paper | | To express HA-NLGN1-His +A1A2 ectodomain |
| Recombinant DNA reagent | pCAGGS-Myc-Amigo | This paper | | To express Myc-Amigo |
| Recombinant DNA reagent | Amigo-CFP | Siddiqui et al., 2010 | | To subclone Myc-Amigo |
| Recombinant DNA reagent | Myc-CD4 | This paper | | To express Myc-CD4 |
| Recombinant DNA reagent | HA-CD4 | Takahashi et al., 2011 | | To subclone Myc-CD4 |
| Recombinant DNA reagent | pLL3.7(hSyn)-Myc-NLGN1*(-A+B)WT | This paper | | To express Myc-NLGN1* -A |
| Recombinant DNA reagent | pLL3.7(hSyn)-Myc-NLGN1*(+A1+B)WT | This paper | | To express Myc-NLGN1* +A1 |
| Recombinant DNA reagent | pLL3.7(hSyn)-CFP-P2A-Myc-NLGN1*(-A+B)P89L | This paper | | To express Myc-NLGN1* -A P89L |
| Recombinant DNA reagent | pLL3.7(hSyn)-CFP-P2A-Myc-NLGN1*(+A1+B)P89L | This paper | | To express Myc-NLGN1* +A1 P89L |

*Continued on next page*

*Continued*

| Reagent type (species) or resource | Designation | Source or reference | Identifiers | Additional information |
|---|---|---|---|---|
| Recombinant DNA reagent | pFB-hSyn-DIO-YFP-P2A-HA-NLGN1*(+A2+B) | *Zhang et al., 2018* | | To subclone NLGN1* isoforms |
| Recombinant DNA reagent | pFB-hSyn-DIO-YFP-P2A-HA-NLGN1*(-A+B) | This paper | | To express Cre-dependent HA-NLGN1* -A |
| Recombinant DNA reagent | pFB-hSyn-DIO-YFP-P2A-HA-NLGN1*(+A1+B) | This paper | | To express Cre-dependent HA-NLGN1* +A1 |
| Recombinant DNA reagent | pCAG-Cre | Addgene | Addgene# 13775 | To express Cre |
| Recombinant DNA reagent | pLL3.7U6-sh*Nlgn1*-hSyn-YFP | *Zhang et al., 2018* | | To express sh*Nlgn1* |
| Recombinant DNA reagent | pLL3.7U6-sh*Nlgn1*-hSyn-CFP | This paper | | To express sh*Nlgn1* |
| Recombinant DNA reagent | pLL3.7(hSyn)-MCS | This paper | | To maintain uniform DNA amounts for transfections |
| Recombinant DNA reagent | pECFP-N1 | Clontech | | To express CFP |
| Recombinant DNA reagent | AP-NLGN1*(-A) | This paper | | To express AP-NLGN1* -A |
| Recombinant DNA reagent | AP-NLGN1*(+A1) | This paper | | To express AP-NLGN1* +A1 |
| Recombinant DNA reagent | sh*Nlgn1*-GFP | *Chamma et al., 2016* | | To express sh*Nlgn1* |
| Recombinant DNA reagent | dsRed-Homer1C | *Mondin et al., 2011* | | To express dsRed-Homer1c |
| Recombinant DNA reagent | BirA$^{ER}$ | *Howarth et al., 2006* | | To express BirA$^{ER}$ |
| Chemical compound, drug | Bicuculline methiodide | Abcam | Cat# ab120109 | |
| Chemical compound, drug | Tetrodotoxin (TTX) | Abcam | Cat# ab120054 | |
| Other | Ni-NTA agarose beads | QIAGEN | Ca# 30210 | |
| Other | Heparin agarose | GE Healthcare | Cat# 17-0406-01 | |
| Software, algorithm | WinLTP | WinLTP | RRID:SCR_008590 | |
| Software, algorithm | MiniAnalysis | Synaptosoft | RRID:SCR_002184 | |
| Software, algorithm | Fiji 64-bit | https://doi.org/10.1038/nmeth.2019 | RRID:SCR_002285 | |
| Software, algorithm | Pymol | Pymol | RRID:SCR_000305 | |
| Software, algorithm | OLego v1.1.2 | *Wu et al., 2013* | RRID:SCR_005811 | |
| Software, algorithm | R | R Foundation | RRID:SCR_001905 | |
| Software, algorithm | GraphPad Prism | GraphPad | RRID:SCR_002798; SCR_000306 | |
| Software, algorithm | GraphPad InStat | GraphPad | RRID:SCR_000306 | |
| Software, algorithm | GraphPad SigmaPlot | GraphPad | RRID:SCR_000306 | |

## DNA constructs

Mouse NLGN1(-A+B) (GenBank: XM_017319496) was cloned from adult mouse whole brain by RT-PCR. Insert A1 (GenBank: XM_017319494.1) and A1A2 (GenBank: XM_006535415.2) were inserted by site-directed mutagenesis. NLGN1(+A2+B) was a gift from Peter Scheiffele (*Scheiffele et al., 2000*). To construct the P89L mutant, nucleotides CCA (Pro) were changed to CTA (Leu). A Myc tag and linker (EQKLISEEDLGGQ) were inserted between amino acid Gln46 and Lys47. CFP-linker-P2A (YGSGATNFSLLKQAGDVEENPGP) was fused at the N-terminus of Myc-NLGN1 before the signal sequence. For the heparin binding assay, since pcDNA4-HA-ecto-NLGN1 (*Zhang et al., 2018*) is the NLGN1(+A2) isoform, the A2 region was removed to construct NLGN1(-A) or replaced with insert A1 or A1A2 for the NLGN1(+A1) or NLGN1(+A1A2) isoform, respectively, to express the NLGN1 ectodomain of each isoform. For coculture assays, Myc-NLGN1 isoforms and CFP-P2A-Myc-NLGN1(-A) WT and P89L mutant were inserted into pCAGGS under a CAG promotor to express in COS7 cells. To construct Myc-Amigo, NLGN1 was replaced with rat Amigo (amino acid 28–493) from Amigo-CFP (*Siddiqui et al., 2010*; *Zhang et al., 2018*). To construct Myc-CD4, the HA tag in HA-CD4 (*Takahashi et al., 2011*) was replaced with the Myc tag. For co-expression of Myc-NLGN1 and CFP in COS7 cells, pECFP-N1 (Clontech) was used. For VGlut1 recruitment assays, Myc-NLGN1 isoforms and CFP-P2A-Myc-NLGN1 (-A) and (+A1) P89L mutant were inserted into pLL3.7 under a synapsin promoter to express in hippocampal neurons. For electrophysiology, NLGN1(-A) and NLGN1 (+A1) isoforms were made from pFB-hSyn-DIO-YFP-P2A-HA-NLGN1, which is the NLGN1(+A2) isoform (*Zhang et al., 2018*). The A2 region was removed for the NLGN1 (-A) isoform or replaced with A1 to construct the NLGN1(+A1) isoform. pCAG-Cre was described previously (*Zhang et al., 2018*). To knock-down endogenous NLGN1, for most experiments pLL3.7U6-sh*Nlgn1*-hSyn-CFP was used which was generated from pLL3.7U6-sh*Nlgn1*-hSyn-YFP and the shRNA resistant form NLGN1* made by mutating the underlined nucleotides 5'-CAAGG<u>GGA</u>A<u>GG</u><u>GTT</u>GAAGTT-3' to 5'-CAAGG<u>C</u>-GA<u>GGG</u>AC<u>T</u>AAAGTT-3' as previously reported (*Zhang et al., 2018*). For the synaptic enrichment assay, the plasmids for sh*Nlgn1*-GFP, RNAi-resistant AP-NLGN1*(+A2+B), BirA[ER], and dsRed-Homer1c were previously described, with thanks to Alice Ting (*Chamma et al., 2016*; *Howarth et al., 2006*; *Mondin et al., 2011*). AP-NLGN1*(-A) and (+A1) were generated from the (+A2) form by In-Fusion HD Cloning (Takara Bio) or site-directed mutagenesis.

## Cell culture and transfection

This study was performed in accordance with the recommendations of the Canadian Council on Animal Care. All of the animals were handled according to approved institutional animal care and use committee (IACUC) protocols (A16-0086 and A16-0090) of the University of British Columbia. For most experiments, primary hippocampal neurons were prepared from male and female rat embryos at embryonic day 18 as previously reported (*Kaech and Banker, 2006*; *Zhang et al., 2018*). The dissociated hippocampal neurons were plated on 18 mm round coverslips coated with poly-L-lysine at the density of 36,000 neurons/coverslip. The coverslips were flipped over and neurons grown suspended above a glial feeder layer in a 60 mm dish. To block glia cell proliferation, cytosine arabinoside at a final concentration of 5 mM was added at day *in vitro* (DIV) 1. Neurons were transfected with 2 μg of plasmid DNA and 2 μl of Lipofectamine 2000 (Thermo Fisher Scientific) at DIV 5. The following are the combinations of plasmids used for each assay with ratio 10:8:2: for VGlut1 immunofluorescence, U6-sh*Nlgn1*-hSyn-CFP + hSyn-Myc-NLGN1* WT + pLL(syn)-MCS (empty vector) or U6-sh*Nlgn1*-hSyn-YFP + hSyn-CFP-P2A-Myc-NLGN1* P89L + pLL(syn)-MCS (empty vector); for electrophysiology, U6-sh*Nlgn1*-hSyn-YFP + hSyn-DIO-YFP-P2A-HA-NLGN1* (-A) or (+A1) + pCAG-Cre.

For the synaptic enrichment assay, dissociated rat hippocampal neurons were electroporated with the Amaxa system (Lonza) using 300,000 cells per cuvette with the following combination of plasmids: sh*Nlgn1*-GFP (3 μg), BirA[ER] (1 μg), Homer1c-dsRed (1 μg), and AP-NLGN1* (-A) or (+A1) (1 μg). Electroporated neurons were immediately resuspended in Minimal Essential Medium supplemented with 10% horse serum and plated on 18 mm coverslips previously coated with poly-L-lysine. Three hours after plating, coverslips were flipped onto 60 mm dishes containing a glial cell feeder layer in Neurobasal Plus Medium supplemented with NeuroCult SM1 Neuronal supplement (STEMCELL Technologies), and cultured for 2 weeks before experiments.

Mycoplasma-negative COS7 cells or HEK239T cells obtained from ATCC without further authentication were maintained in Dulbecco's Modified Eagle's medium (Thermo Fisher Scientific) with 10%

bovine growth serum. For transfection, plasmid DNA and 0.045% Polyethylenimine (Sigma 408727) were mixed in a 1:3 or 1:4 ratio (w/v). For coculture assays, an N-methyl-D-aspartate (NMDA) receptor inhibitor, APV, at a final concentration of 100 µM was added into the neuron dishes at DIV 6, 9, and 13. COS7 cells were trypsinized 24 hr after transfection. 40,000 COS7 cells were seeded on a coverslip with neurons at DIV 13. After 1 hr, the coverslip was flipped back into a glia dish and 24 hr later, the COS7-neuron cocultures were fixed.

## Immunocytochemistry

For coculture assays, neurons and COS7 cells were fixed with 4% formaldehyde/4% sucrose/PBS, pH 7.4 for 12 min, blocked with 10% BSA/PBS for 30 min at 37°C, and incubated with Myc antibody in 3% BSA/5% normal goat serum (NGS)/PBS for surface staining overnight at 4°C. The next day, the cells were fixed again for 5 min, permeabilized with 0.2% Triton X-100/PBS, blocked with 10% BSA, and incubated with the primary antibodies in 3% BSA/5% NGS/PBS overnight at 4°C. Cells were then incubated with the secondary antibodies in 3% BSA/5% NGS/PBS for 45 min at 37°C. The coverslips were mounted in elvanol (Tris-HCl, glycerol, and polyvinyl alcohol, with 2% 1,4-diazabicyclo {2,2,2} octane). For surface expression of Myc-NLGN1(-A) WT and P89L in COS7 cells, cells were fixed and blocked as above 48 hr after transfection, followed by incubation with Myc antibody overnight at 4°C, and incubation with the secondary antibody the next day. For VGlut1 immunofluorescence assays, the transfected hippocampal neurons were fixed at DIV 12 and immunostained as above, except the second 5 min fixation was not done. The following primary antibodies were used: Rabbit polyclonal Myc antibody (Sigma C3956; 1:40000), rabbit polyclonal pan-NRXN antibody (Millipore ABN161; 1:1000), chicken polyclonal MAP2 antibody (Abcam ab5392; 1:80000), mouse monoclonal VGlut1 antibody (NeuroMab 75–066; clone N28/9; IgG1; 1:4000), mouse monoclonal SynapsinI antibody (Synaptic Systems 106 011; clone 46.1; IgG1; 1:20000), mouse monoclonal Tau antibody (Millipore MAB3420; clone PC1C6; IgG2a; 1:2000). Secondary antibodies coupled to Alexa Flour 488, 568, 647 (Thermo Fisher Scientific; 1: 1000) and anti-chicken IgY AMCA (Jackson ImmunoResearch 703-155-155; 1:400) were used. Image acquisition and analysis was performed blind to the experimental condition. Images were acquired on a Zeiss Axioplan 2 microscope with 63x/1.4 numerical aperture (NA) oil objective, ORCA-Flash4.0 FL CMOS camera (Hamamatsu) and custom filter sets or a Zeiss LSM700 with 40x/1.4 NA oil objective, ORCA-Flash4.0 CMOS camera (Hamamatsu) and custom filter sets. The acquired images were analysed with Fiji. Off-cell background was subtracted from each value. COS7 cells or neurons were selected for analysis based only on cell health and cell surface expression level of Myc-NLGN1, except for *Figure 8—figure supplement 1* where cells were chosen based on CFP expression. For the coculture assays, the total intensity of pan-NRXN or Synapsin puncta was measured within the area of COS7 cells where Tau-positive axon was present and MAP2-positive dendrite was absent. For the neuron cultures, the number and total intensity of VGlut1 puncta was measured and normalized to the area or length of MAP2-positive dendrite.

For the synaptic enrichment assay, for NLGN1 staining, cell cultures were rinsed twice in pre-warmed Tyrode solution (15 mM D-glucose, 108 mM NaCl, 5 mM KCl, 2 mM MgCl$_2$, 2 mM CaCl$_2$ and 25 mM HEPES, pH 7.4, 280 mOsm). Cells were then incubated for 10 min at 37°C with Tyrode solution containing 1% biotin-free Bovine Serum Albumin (Roth, ref 0163.4), followed by an incubation in 100 nM Alexa 647-conjugated streptavidin (cat#S32357, Thermo Fisher Scientific). Following two washes in Tyrode solution, cells were imaged on an inverted epifluorescence microscope (Nikon Eclipse TiE) equipped with a 60x/1.49 NA objective lens and an sCMOS camera (Prime 95B, Photometrics) driven by the Metamorph software (Molecular Devices). Homer1c-DsRed puncta were thresholded using Metamorph, and the number of puncta per dendrite length was determined, as well as area and intensity relative to dendrite shaft. Regions around Homer1c-DsRed objects were created and transferred to the streptavidin image. The intensity of the streptavidin signal in the puncta was normalized to the shaft signal, giving the NLGN1 synaptic enrichment.

## Heparin binding assay

Purification of NLGN1 ectodomain proteins and the heparin binding assay were previously described (*Zhang et al., 2018*). Briefly, an N-terminal HA tagged and C-terminal His tagged NLGN1 ectodomain of each isoform was transiently expressed in HEK293T cells and purified with Ni-NTA agarose

beads (QIAGEN). Purified NLGN1 ectodomains were incubated with heparin agarose (GE Health-care) and eluted with NaCl in step-wise gradient concentrations (50 mM – 3 M) in 20 mM HEPES, pH 7.4. Collected eluates were analysed by immunoblotting with rat monoclonal HA antibody (Roche 11867431001; clone 3F10; 1:2000), secondary HRP-conjugated antibody, and detected using the SuperSignal Chemiluminescent kit (Thermo Fisher Scientific).

## Electrophysiology

Cultured hippocampal neurons (DIV 13–14) were transferred to a submerged chamber and continuously perfused (2–3 ml/min) with (in mM): 140 NaCl, 5.4 KCl, 1.25, 1 $MgCl_2$, 1.3 $CaCl_2$, 10 HEPES and 3.6 D-glucose with pH adjusted to 7.35 using NaOH. All chemicals and drugs were purchased from Sigma or BioShop, Canada unless otherwise stated. YFP transfected neurons were identified with a microscope equipped with fluorescent and phase contrast optics. All recordings were performed at room temperature with a MultiClamp 700B amplifier. Cells were voltage clamped at −60 mV. Recording of mEPSCs was conducted using pipettes filled with an intracellular solution containing (in mM): 122.5 Cs-methanesulfonate, 17.5 CsCl, 2 $MgCl_2$, 10 EGTA, 10 HEPES, 4 ATP (K), and 5 QX-314, with pH adjusted to 7.2 by CsOH. Tetrodotoxin (500 nM; Abcam) and bicuculline methiodide (10 µM; Abcam) were added to perfused saline to block action potentials and GABA receptor-mediated inhibitory synaptic currents, respectively. mEPSCs were recorded using WinLTP software in continuous acquisition mode. Analyses for frequency and amplitude were conducted using MiniAnalysis software.

## Analysis of RNA-seq data and quantification of NLGN1 isoform abundance

All RNA-seq datasets analysed in this work, including a brief description of samples, read length, sequencing depth, and accession numbers are summarized in *Supplementary file 1*. In particular, neuronal cell type-specific splicing of NLGN1 was analysed using single cell RNA-seq data of adult mouse cortex, which is composed of ~25,000 cells of diverse neuronal cell types from primary visual cortex (VISp) and anterior lateral motor cortex (ALM) (*Tasic et al., 2018*). In our analysis, we used ~21000 core cells unambiguously assigned to specific neuronal cell types defined by transcriptional profiles in the original study. In addition, we also examined NLGN1 splicing in major glutamatergic and GABAergic neuronal cell types in the cortex and hippocampus derived by translational profiling (*Furlanis et al., 2019*). For each of these datasets, RNA-seq reads were mapped by OLego v1.1.2 (*Wu et al., 2013*) to the reference genome (mm10) and a comprehensive database of exon junctions was provided for read mapping. Only reads unambiguously mapped to the genome or exon junctions (single hits) were used for downstream analysis. RNA-seq data for cortical development and RNA-binding protein depletion were processed and analysed previously (*Jacko et al., 2018*; *Weyn-Vanhentenryck et al., 2018*; *Yan et al., 2015*) and the number of reads mapped to each exon and exon junctions derived from the original analysis was used to quantify NLGN1 isoforms.

To estimate the abundance of NLGN1 isoforms, we first extracted the number of reads covering each of the six exon junctions connecting the alternatively spliced A1 and A2 exons and their flanking constitutive exons. The read counts from replicates in bulk data or from single cells assigned to the same cell types in the single cell RNA-seq data were pooled together. The read count of the six junctions were then taken as input to estimate the fraction of the four NLGN1 splice isoforms including A1A2, A1, A2 and –A using least squares estimation with the R function lsfit.

## Statistical analysis

Statistical analysis was performed using Microsoft Excel, GraphPad Prism, GraphPad InStat, and SigmaPlot. Statistical details are provided in the Figure legends. For most experiments, data did not pass the D'Agostino and Pearson test for normality so comparisons were made using the Mann-Whitney test or Kruskal-Wallis test with post hoc Dunn's multiple comparisons tests, as indicated. For the VGlut1 analysis, an outlier was removed using an aggressive Grubbs' method $\alpha$ = 0.0001 in GraphPad Prism. All data are reported as the mean ± standard error of the mean (SEM).

## Acknowledgements

We thank Xiling Zhou and Anny Chih for assistance with neuron cultures and fluorescence measures. This work was supported by Canadian Institutes of Health Research FDN-143206 and Simons Foundation Autism Research Initiative SFARI 608066 grants to AMC, National Institutes of Health R01NS089676 and R01GM124486 grants to CZ, and Agence Nationale pour la Recherche (grant « Synthesyn » ANR-17-CE16-0028-01) and Fondation pour la Recherche Médicale (« Equipe FRM » DEQ20160334916) to OT.

## Additional information

### Funding

| Funder | Grant reference number | Author |
| --- | --- | --- |
| Simons Foundation | SFARI 608066 | Ann Marie Craig |
| Canadian Institutes of Health Research | FDN-143206 | Ann Marie Craig |
| National Institutes of Health | R01NS089676 | Chaolin Zhang |
| National Institutes of Health | R01GM124486 | Chaolin Zhang |
| Agence Nationale de la Recherche | grant « Synthesyn » ANR-17-CE16-0028-01 | Olivier Thoumine |
| Fondation pour la Recherche Médicale | « Equipe FRM » DEQ20160334916 | Olivier Thoumine |

The funders had no role in study design, data collection and interpretation, or the decision to submit the work for publication.

### Author contributions

Shinichiro Oku, Conceptualization, Formal analysis, Validation, Investigation, Visualization, Methodology, Writing - original draft, Writing - review and editing; Huijuan Feng, Software, Formal analysis, Visualization, Writing - review and editing, RNA-seq analyses; Steven Connor, Formal analysis, Investigation, Visualization, Methodology, Writing - review and editing, electrophysiological analyses; Andrea Toledo, Formal analysis, Investigation, Visualization, Methodology, Writing - review and editing, synaptic enrichment and Homer cluster analyses; Peng Zhang, Formal analysis, Investigation, Visualization, Methodology, Writing - review and editing, heparin binding; Yue Zhang, Formal analysis, Validation, coculture image analyses; Olivier Thoumine, Resources, Supervision, Funding acquisition, Validation, Visualization, Project administration, Writing - review and editing, synaptic enrichment and Homer cluster analyses; Chaolin Zhang, Resources, Supervision, Funding acquisition, Validation, Visualization, Project administration, Writing - review and editing, RNA-seq analyses; Ann Marie Craig, Conceptualization, Resources, Formal analysis, Supervision, Funding acquisition, Validation, Visualization, Writing - original draft, Project administration, Writing - review and editing

### Author ORCIDs

Shinichiro Oku https://orcid.org/0000-0002-1916-1870
Olivier Thoumine http://orcid.org/0000-0002-8041-1349
Chaolin Zhang http://orcid.org/0000-0002-8310-7537
Ann Marie Craig https://orcid.org/0000-0002-8651-8200

### Ethics

Animal experimentation: This study was performed in accordance with the recommendations of the Canadian Council on Animal Care. All of the animals were handled according to approved institutional animal care and use committee (IACUC) protocols (A16-0086 and A16-0090) of the University of British Columbia.

**Decision letter and Author response**
Decision letter https://doi.org/10.7554/eLife.58668.sa1
Author response https://doi.org/10.7554/eLife.58668.sa2

# Additional files

## Supplementary files

• Supplementary file 1. Summary of RNA-seq datasets analysed in this study. Accession numbers refer to the Gene Expression Omnibus or Sequence Read Archive. Sample description ages indicate embryonic days (E) or postnatal days (P) or weeks (W). Genotypes refer to single (KO), double (DKO) and triple (TKO) knockout ESC-derived neurons or mice. Library type indicates paired end (PE) or single end (SE).

• Transparent reporting form

## Data availability

All data generated during this study are included in the manuscript and supporting files. Source data files have been provided for Figures 2, 5 and 8.

The following previously published datasets were used:

| Author(s) | Year | Dataset title | Dataset URL | Database and Identifier |
|---|---|---|---|---|
| Tasic B, Yao Z, Graybuck LT, Smith KA, Nguyen TN, Bertagnolli D, Goldy J, Garren E, Economo MN, Viswanathan S, Penn O, Bakken T, Menon V, Miller J, Fong O, Hirokawa KE, Lathia K, Rimorin C, Tieu M, Larsen R, Casper T, Barkan E, Kroll M, Parry S, Shapovalova NV, Hirschstein D, Pendergraft J, Sullivan HA, Kim TK, Szafer A, Dee N, Groblewski P, Wickersham I, Cetin A, Harris JA, Levi BP, Sunkin SM, Madisen L, Daigle TL, Looger L, Bernard A, Phillips J, Lein E, Hawrylycz M, Svoboda K, Jones AR, Koch C, Zeng H | 2018 | Shared and distinct transcriptomic cell types across neocortical areas | https://www.ncbi.nlm.nih.gov/geo/query/acc.cgi?acc=GSE115746 | NCBI Gene Expression Omnibus, GSE115746 |
| Furlanis E, Traunmuller L, Fucile G, Scheiffele P | 2019 | Landscape of ribosome-engaged transcript isoforms reveals extensive neuronal-cell-class-specific alternative splicing programs | https://www.ncbi.nlm.nih.gov/geo/query/acc.cgi?acc=GSE133291 | NCBI Gene Expression Omnibus, GSE133291 |
| Charizanis K, Lee KY, Batra R, Goodwin M, Zhang C, Yuan Y, Shiue L, Cline M, Scotti MM, Xia G, Kumar A, Ashizawa T, Clark HB, Kimura T, Ta- | 2012 | Muscleblind-like 2-mediated alternative splicing in the developing brain and dysregulation in myotonic dystrophy | https://www.ncbi.nlm.nih.gov/geo/query/acc.cgi?acc=GSE38497 | NCBI Gene Expression Omnibus, GSE38497 |

| | | | | |
|---|---|---|---|---|
| kahashi MP, Fuji-mura H, Jinnai K, Yoshikawa H, Gomes-Pereira M, Gourdon G, Sakai N, Nishino S, Foster TC, Ares M, Darnell RB, Swanson MS | | | | |
| Lister R, Mukamel EA, Nery JR, Urich M, Puddifoot CA, Johnson ND, Lucero JD, Huang Y, Dwork AJ, Schultz MD, Yu M, Tonti-Filippini J, Heyn H, Hu S, Wu JC, Rao A, Esteller M, He C, Haghighi FG, Sejnowski TJ, Behrens MM, Ecker JR | 2013 | Global epigenomic reconfiguration during mammalian brain development | https://www.ncbi.nlm.nih.gov/geo/query/acc.cgi?acc=https://www.ncbi.nlm.nih.gov/geo/query/acc.cgi?acc=GSE47966 | NCBI Gene Expression Omnibus, GSE47966 |
| Yan Q, Weyn-Vanhentenryck SM, Wu J, Sloan SA, Zhang Y, Chen K, Wu JQ, Barres BA, Zhang C | 2015 | Systematic discovery of regulated and conserved alternative exons in the mammalian brain reveals NMD modulating chromatin regulators | https://www.ncbi.nlm.nih.gov/sra/?term=SRP055008 | NCBI Sequence Read Archive, SRP055008 |
| Li Q, Zheng S, Han A, Lin CH, Stoilov P, Fu XD, Black DL | 2014 | The splicing regulator PTBP2 controls a program of embryonic splicing required for neuronal maturation. | https://www.ncbi.nlm.nih.gov/geo/query/acc.cgi?acc=GSE51733 | NCBI Gene Expression Omnibus, 51733 |
| Weyn-Vanhentenryck SM, Feng H, Ustianenko D, Duffie R, Yan Q, Jacko M, Martinez JC, Goodwin M, Zhang X, Hengst U, Lomvardas S, Swanson MS, Zhang C | 2018 | Precise temporal regulation of alternative splicing during neural development | https://www.ncbi.nlm.nih.gov/sra/?term=SRP142522 | NCBI Sequence Read Archive, SRP142522 |
| Jacko M, Weyn-Vanhentenryck SM, Smerdon JW, Yan R, Feng H, Williams DJ, Pai J, Xu K, Wichterle H, Zhang C | 2018 | Rbfox Splicing Factors Promote Neuronal Maturation and Axon Initial Segment Assembly | https://www.ncbi.nlm.nih.gov/sra/?term=SRP128054 | NCBI Sequence Read Archive, SRP128054 |

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
