## [Decision Letter]

**Acceptance summary:**

The authors show that isoforms of Neuroligin1 that include the A1 alternatively-spliced insert bind more tightly to heparan sulfate which is a post-translational modification of neurexin. They show that the A1 insert is most common in GABAergic neurons in the forebrain. The findings support the notion that inclusion of the A1 insert increases binding affinity for neurexin and this increases the function of neuroligin1 as an organizer of presynaptic terminals at the site of developing synapses. They show that inclusion of the A1 insert in a mutant form of neuroligin1 that is associated with autism spectrum disorders enhances its function, suggesting that therapies that could increase inclusion of the A1 insert might be "partially efficacious" in autistic children with neurexin mutations.

**Decision letter after peer review:**

Thank you for submitting your article "Alternative Splicing at Neuroligin Site A Regulates Glycan Interaction and Synaptogenic Activity" for consideration by *eLife*. Your article has been reviewed by three peer reviewers, including Mary B Kennedy as the Reviewing Editor and Reviewer #1, and the evaluation has been overseen by Marianne Bronner as the Senior Editor.

The reviewers have discussed the reviews with one another and the Reviewing Editor has drafted this decision to help you prepare a revised submission.

We would like to draw your attention to changes in our revision policy that we have made in response to COVID-19 (https://elifesciences.org/articles/57162). Specifically, we are asking editors to accept without delay manuscripts, like yours, that they judge can stand as *eLife* papers without additional data, even if they feel that they would make the manuscript stronger. Thus the revisions requested below mainly address clarity and presentation. We also include several suggestions for additional experiments that you might consider for a Research Advance article.

Summary:

The manuscript by Oku et al. on alternative splicing at Neuroligin site A is an interesting contribution to our understanding of regulation of synapse formation by neuroligin. While there are many studies on NLGN1 splice site B, little is known about splice site A. The authors show that isoforms of Neuroligin1 that include the A1 alternatively-spliced insert bind more tightly to heparin sulfate which is a post-translational modification of neurexin. The authors used published deep RNA-seq datasets from neuronal cells to show that the A1 insert is differentially expressed, being more highly expressed in GABAergic cell types and certain oligodendrocytes in the visual cortex and anterior lateral motor cortex. This finding was corroborated by analysis of an additional cortical datasets from a different laboratory group. Using RNA-seq datasets from Rbfox knockout mice, the authors show that knockout of all three isoforms, Rbfox1, Rbfox2, and Rbfox3, eliminates inclusion of the A1 insert. They use a well-established tissue co-culture assay to show that the presence of the A1 insert increases neurexin recruitment to surface neuroligin, consistent with higher affinity for neurexin. They show that the presence of the A1 insert increases presynaptic differentiation at the site of recruitment, and synaptic transmission (i.e. mini-frequency) at these sites. All of these findings support the notion that the alternatively spliced A1 insert in neuroligin1 is differentially expressed and its presence increases its recruitment of neurexin and increases its function as an organizer of presynaptic terminals at the site of developing post-synapses. The analysis of transcripts uncovered a prominent role for Rbfox in increasing the inclusion of the A1 insert. Finally, the authors showed that inclusion of the A1 insert in a mutant form of neuroligin1 that is associated with ASD, enhanced its function, suggesting that therapies that could increase inclusion of the A1 insert might be "partially efficacious" in ASD children with neurexin mutations.

Essential revisions:

The reviewers request several revisions, mostly concerning the clarity of presentation, and one request for an additional experiment if possible within 2 months. Otherwise, further discussion of the results should be added.

1) Some of the reviewers considered that there was a mechanistic disconnect between the first figure, showing increased affinity for heparin sulfate when the A1 insert is present, and the rest of the findings. For example: "The authors demonstrate that the positively charged A1 insert increases the affinity for heparin and propose that the A1 insert promotes interaction with heparin sulfate (HS) on Nrxn. Indeed the NL1 + A1 construct shows enhanced synaptogenic activity in the coculture assay. However, NL1 can interact with the Nrxn LNS domain as well as its HS moiety. Determining the binding affinity of NL1 – A1 and + A1 for Nrxns to test whether the A1 insert increases affinity for Nrxn would therefore strengthen this claim. SPR data in Comoletti et al., 2006 suggests that the presence of A1 indeed increases affinity for Nrxn1β, but it would be helpful to confirm this with the constructs used in this study. On the other hand, a previous paper from the Craig lab (Elegheert et al., 2017 showed no effect of the A1 insert on affinity for neurexin in an SPR study." The reviewers would like to see an experiment in which the affinity of neuroligin, plus and minus the A1 insert, for neurexin on neurons is tested more directly. This could be an experiment similar to the one published from your lab in the Zhang et al., 2018 paper in Figure 3I, J. In the first paragraph of the present manuscript under the heading "The NL1 A1 insert enhances presynaptic differentiation in coculture" you state, "If the observed differences in heparin binding among NL splice variants (Figure 1) reflect differences in binding to native HS modified Nrx, we expect NL1 A1-containing splice variants to recruit more Nrx than NL1 variants lacking A1. Indeed, we found that Myc-tagged NL1 +A1 and +A1A2 splice forms expressed on COS7 cells cocultured with rat hippocampal neurons recruited more native neuronal Nrx to contact sites than NL1 -A or +A2 (Figure 4A-C)." If the requested direct binding experiment is not available within 2 months, you should move the presentation of this data to a separate paragraph under a distinct heading stating that the A1 insert appears to enhance binding of neuroligin on heterologous cells to native neurexin in neurons. It might be a good idea to move this paragraph and Figure 4 to a Figure 2 immediately following Figure 1. In the paragraph, please explain why the findings in Ko et al., 2009 support this statement. There was concern among the reviewers about whether neurexin is required for heterologous synapse formation in the co-culture experiment. Also, if the experiment is not available, in the Discussion, please mention further experiments that would clarify whether the A1 insert increases affinity for native neurexin. You might consider such experiments, and some of the other suggestions below, as a "Research Advance" tied to this paper in the future.

2) The description of the analyses of online RNA-seq datasets in the Results section did not do justice to the significance of the authors' conclusions. It is important that readers be able to evaluate the size and appropriateness of the data on which the analyses are based, without having to look up each paper. For each dataset analyzed, the authors should include a description of the size of the dataset (from the published papers) and the variables that were examined (e.g. cell types), the conclusions that were drawn from the datasets when they were published, the new information that the authors sought by re-analyzing the data (e.g. Is there differential expression of the A1 splice variant?), and then the conclusions reached from the authors' analysis (e.g. The A1 variant is more highly expressed in GABAergic neurons sampled by the datasets.) The developmental dataset should be described in a separate paragraph to avoid confusion about the origin of the developmental data. We note that the data analyzed in Figure 3B shows a peak in the second postnatal week rather than the first postnatal week. Thus, the summary statement in the Results should address the slight discrepancy between datasets in Figure 3A and B. Similarly, for Figures 3C, D, and E, each data set should be described and the meaning of the Cre expressing constructs in the bar labels should be explained.

3) Figure 4: The authors showed in Figure 1 that the insert A1 and insert A1A2 have the same affinity for heparin. In Figure 4B, it appears that + A1A2 has a bigger effect on the neurexin signal than + A1. Is this comparison significant?

4) Figures 7 and 8: Nakanishi et al. previously showed that NLGN1 P89L does not express robustly at the surface in COS7 cells, and P89L led to a decrease in spine number. Here, the authors show that NLGN P89L is expressing at the surface as well as WT NLGN1. Can the authors provide more information about this? Nakanishi et al. reported that NLGN1 P89L had a deficit in spine number when overexpressed in hippocampal neurons. The authors showed here that + A1 increased Vglut intensity, but from the image, it also appeared that there are more VGlut puncta in NLGN1 P89L + A1. The authors showed previously that Vglut1 clustering data were included. Can the authors also provide that dataset and quantification here. Lastly, NLGN1 WT should be included in this experiment to show if NLGN1 P89L + A1 can restore to a comparable level.

5) Additional issues raised by the reviewers that might be considered in a research advance submission:

Figure 6: The authors hypothesize that the increase in mEPSC frequency is due to an increase in presynaptic differentiation due to an increase in VGlut intensity form the ICC experiment. The inclusion of a PPE experiment to determine if the changes are due to presynaptic and not postsynaptic effects would be beneficial.

Figure 5: How does NL1 A1+ enhance Vglut1 clustering? Is this effect Nrxn dependent? Does it depend on the presence of HS in Nrxn?

Related to the previous comment, a retrograde effect of NL1 on presynaptic vesicle recruitment was previously shown to require N-Cadherin (Stan et al., 2010). Does the insertion of A1 affect NL1-Cadherin interactions?

6) Please indicate in numbering which figure the supplementary figure should be associated with per *eLife* policy (see requested format online). We assume that is Figure 1.

---

## [Author Response]

Essential revisions:The reviewers request several revisions, mostly concerning the clarity of presentation, and one request for an additional experiment if possible within 2 months. Otherwise, further discussion of the results should be added.1) Some of the reviewers considered that there was a mechanistic disconnect between the first figure, showing increased affinity for heparin sulfate when the A1 insert is present, and the rest of the findings. For example: "The authors demonstrate that the positively charged A1 insert increases the affinity for heparin and propose that the A1 insert promotes interaction with heparin sulfate (HS) on Nrxn. Indeed the NL1 + A1 construct shows enhanced synaptogenic activity in the coculture assay. However, NL1 can interact with the Nrxn LNS domain as well as its HS moiety. Determining the binding affinity of NL1 – A1 and + A1 for Nrxns to test whether the A1 insert increases affinity for Nrxn would therefore strengthen this claim. SPR data in Comoletti et al., 2006 suggests that the presence of A1 indeed increases affinity for Nrxn1β, but it would be helpful to confirm this with the constructs used in this study. On the other hand, a previous paper from the Craig lab (Elegheert et al., 2017 showed no effect of the A1 insert on affinity for neurexin in an SPR study."

Thank you for raising this important point. These previous studies assessing Nrx‐NL affinity by SPR used recombinant Nrx constructs lacking the HS modification site. Thus, the effect of the NL1 A1 insert on binding to the HS modification of Nrx would not have been tested in these studies. Further, even the full ectodomain of Nrx produced in HEK293 cells is poorly HS modified, making such direct binding studies difficult. These issues are now explained in the second paragraph of the Results: “We used this cell-based assay to assess Nrx‐NL1 interaction because recombinant Nrx produced in HEK293 cells is poorly HS modified (Zhang et al., 2018), making it difficult to do direct binding assays with appropriately glycosylated Nrx. […] Altogether, these previous results and our data suggest that the NL1 A1 insert does not affect binding to the Nrx LNS domain but increases binding to the Nrx HS modification.”

The reviewers would like to see an experiment in which the affinity of neuroligin, plus and minus the A1 insert, for neurexin on neurons is tested more directly. This could be an experiment similar to the one published from your lab in the Zhang et al., 2018 paper in Figure 3I, J.

We agree that this is a good idea. We tried to do this experiment but were unable to complete it during the 2‐month window for revision with restricted lab access. We had problems generating good preparations of HA‐NL1‐His ‐A and +A1 ectodomain proteins on time to generate binding curves with the one primary neuron culture we were able to generate during this period.

In the first paragraph of the present manuscript under the heading "The NL1 A1 insert enhances presynaptic differentiation in coculture" you state, "If the observed differences in heparin binding among NL splice variants (Figure 1) reflect differences in binding to native HS modified Nrx, we expect NL1 A1-containing splice variants to recruit more Nrx than NL1 variants lacking A1. Indeed, we found that Myc-tagged NL1 +A1 and +A1A2 splice forms expressed on COS7 cells cocultured with rat hippocampal neurons recruited more native neuronal Nrx to contact sites than NL1 -A or +A2 (Figure 4A-C)." If the requested direct binding experiment is not available within 2 months, you should move the presentation of this data to a separate paragraph under a distinct heading stating that the A1 insert appears to enhance binding of neuroligin on heterologous cells to native neurexin in neurons. It might be a good idea to move this paragraph and Figure 4 to a Figure 2 immediately following Figure 1. In the paragraph, please explain why the findings in Ko et al., 2009 support this statement. There was concern among the reviewers about whether neurexin is required for heterologous synapse formation in the co-culture experiment.

Indeed, we appreciate that this experiment addresses more the interaction of Nrx and NL1 rather than functional consequences and fits better earlier in the manuscript. We thus moved the presentation of this data to current Figure 2. There is considerable evidence that neurexin is required for heterologous synapse formation in the co‐culture experiment, from Ko et al., 2009, and other publications. We now provide a supporting statement in this paragraph: “NL binding to axonal Nrx is required for this synaptogenic activity in coculture, based on loss of activity upon Nrx triple knockdown (Gokce and Sudhof, 2013; Zhang et al., 2018) or by NL point mutations that disrupt binding to the Nrx LNS domain (Ko et al., 2009) or to the Nrx HS modification (Zhang et al., 2018).”

Also, if the experiment is not available, in the Discussion, please mention further experiments that would clarify whether the A1 insert increases affinity for native neurexin. You might consider such experiments, and some of the other suggestions below, as a "Research Advance" tied to this paper in the future.

We added the following sentence to the first paragraph of the Discussion: “It will be important in future studies to directly compare the affinities of purified NL1 splice forms with native HS‐modified Nrx, such as with a neuron‐based binding assay.”

2) The description of the analyses of online RNA-seq datasets in the Results section did not do justice to the significance of the authors' conclusions. It is important that readers be able to evaluate the size and appropriateness of the data on which the analyses are based, without having to look up each paper. For each dataset analyzed, the authors should include a description of the size of the dataset (from the published papers) and the variables that were examined (e.g. cell types), the conclusions that were drawn from the datasets when they were published, the new information that the authors sought by re-analyzing the data (e.g. Is there differential expression of the A1 splice variant?), and then the conclusions reached from the authors' analysis (e.g. The A1 variant is more highly expressed in GABAergic neurons sampled by the datasets.) The developmental dataset should be described in a separate paragraph to avoid confusion about the origin of the developmental data. We note that the data analyzed in Figure 3B shows a peak in the second postnatal week rather than the first postnatal week. Thus, the summary statement in the Results should address the slight discrepancy between datasets in Figure 3A and 3B. Similarly, for Figures 3C, D, and E, each data set should be described and the meaning of the Cre expressing constructs in the bar labels should be explained.

We appreciate these comments and greatly expanded the description of the RNA‐seq analyses in the main text, better describing the datasets and clarifying what had been done previously and the new analyses. The description of the developmental data is now presented in its own paragraph and more accurately reports a peak in the first to second postnatal weeks. The RNA‐seq analyses from KO mice for RNA binding proteins is now presented in a separate section. The Cre line KO mouse lines are also now fully explained in the main text and figure legends. Furthermore, we added a new Supplementary file 1 summarizing all the RNA‐seq datasets listing references, accession numbers, samples sources, read length, and sequencing depth so readers can appreciate the size and appropriateness of the data.

3) Figure 4: The authors showed in Figure 1 that the insert A1 and insert A1A2 have the same affinity for heparin. In Figure 4B, it appears that + A1A2 has a bigger effect on the neurexin signal than + A1. Is this comparison significant?

This difference in neurexin recruitment is not significant, as now stated in the figure legend: “Although the value for NL1 +A1A2 was higher than that for NL1 +A1, this difference was not significant.” (current Figure 2).

4) Figures 7 and 8: Nakanishi et al. previously showed that NLGN1 P89L does not express robustly at the surface in COS7 cells, and P89L led to a decrease in spine number. Here, the authors show that NLGN P89L is expressing at the surface as well as WT NLGN1. Can the authors provide more information about this?

We apologize for generating confusion about this point. In our original experiments, we did not assay surface expression, but rather chose cells with equal surface expression, revealing additional deficits in function caused by the P89L mutation beyond any deficit in surface expression. We assumed based on Nakanishi et al. that there is also some deficit in both total and surface levels of P89L NL1 relative to WT, although it was easy to find cells with equal surface expression for P89L and WT NL1. We did new experiments to assess surface expression in COS7 cells, the cell line used in our co‐culture experiments and used by Nakanishi et al., and indeed found a significant reduction in NL1 P89L surface expression relative to WT (new Figure 8—figure supplement 1.) In this assay, we co‐transfected a CFP expression vector along with the Myc‐NL1 WT or P89L expression vectors and chose cells by CFP expression. However, we did not do this in the other assays but rather chose cells by Myc‐NL1 surface expression. We altered the text of the Results and current Figure 8 and 9 legends to clarify this issue, that cells were chosen for equal surface Myc‐NL1 and thus functional differences observed are independent of surface level.

Nakanishi et al. reported that NLGN1 P89L had a deficit in spine number when overexpressed in hippocampal neurons. The authors showed here that + A1 increased Vglut intensity, but from the image, it also appeared that there are more VGlut puncta in NLGN1 P89L + A1. The authors showed previously that Vglut1 clustering data were included. Can the authors also provide that dataset and quantification here. Lastly, NLGN1 WT should be included in this experiment to show if NLGN1 P89L + A1 can restore to a comparable level.

We measured the VGlut1 cluster density for this experiment and found no difference between NL1 P89L ‐A and +A1, only a difference in VGlut1 total intensity, as shown in current Figure 9. Sorry, we did not include NL1 ‐A WT in this experiment, and did not have sufficient lab access to re‐do the experiment during the time window for revision. Based on the co‐culture data (Figure 8), we would only expect a partial restoration towards NL1 ‐A WT by A1 splice inclusion in NL1 P89L.

5) Additional issues raised by the reviewers that might be considered in a research advance submission:Figure 6: The authors hypothesize that the increase in mEPSC frequency is due to an increase in presynaptic differentiation due to an increase in VGlut intensity form the ICC experiment. The inclusion of a PPE experiment to determine if the changes are due to presynaptic and not postsynaptic effects would be beneficial.Figure 5: How does NL1 A1+ enhance Vglut1 clustering? Is this effect Nrxn dependent? Does it depend on the presence of HS in Nrxn?Related to the previous comment, a retrograde effect of NL1 on presynaptic vesicle recruitment was previously shown to require N-Cadherin (Stan et al., 2010). Does the insertion of A1 affect NL1-Cadherin interactions?

We appreciate these suggestions. A paired pulse experiment to assess presynaptic release, testing the effect of the NL1 A1 insert in conditions of neurexin TKD and expression of an HS‐deficient form, and testing the role of N‐cadherin are all good ideas. We will try to address at least some of these issues with a future research advance. We are grateful that such experiments are not required now given our limited lab access.

6) Please indicate in numbering which figure the supplementary figure should be associated with per eLife policy (see requested format online). We assume that is Figure 1.

Yes, this supplementary figure is now designated as Figure 1—figure supplement 1.